# United Minds or Isolated Agents? Exploring Coordination of LLMs under Cognitive Load Theory

## Abstract

Large Language Models (LLMs) exhibit a notable performance ceiling on complex, multi-faceted tasks, as they often fail to integrate diverse information or adhere to multiple constraints. We posit that such limitation arises when the demands of a task exceed the LLM's effective cognitive load capacity. This interpretation draws a strong analogy to Cognitive Load Theory (CLT) in cognitive science, which explains similar performance boundaries in the human mind, and is further supported by emerging evidence that reveals LLMs have bounded working memory characteristics. Building upon this CLT-grounded understanding, we introduce ***CoThinker***, a novel LLM-based multi-agent framework designed to mitigate cognitive overload and enhance collaborative problem-solving abilities. ***CoThinker*** operationalizes CLT principles by distributing intrinsic cognitive load through agent specialization and managing transactional load via structured communication and a collective working memory. We empirically validate *CoThinker* on complex problem-solving tasks and fabricated high cognitive load scenarios, demonstrating improvements over existing multi-agent baselines in solution quality and efficiency. Our analysis reveals characteristic interaction patterns, providing insights into the emergence of collective cognition and effective load management, thus offering a principled approach to overcoming LLM performance ceilings.

## 1 Introduction

The increasing prevalence and capability of Large Language Models (LLMs) are transforming diverse domains, moving beyond basic text generation towards complex reasoning and problem-solving applications [Chang et al., 2024, Zhao et al., 2024, Li et al., 2024a]. Aligning these powerful models with human intent and fostering effective thinking pattern is paramount for unlocking their full potential [Shen et al., 2023]. In-Context Learning (ICL) is increasingly employed for alignment, offering adaptation via prompts without parameter updates [Brown et al., 2020]. In this work, we adopt a broad definition of ICL, referring to the general strategy of guiding an LLM's behavior by providing any contextual information relevant to the task to perform the task [Lampinen et al., 2024]. Compared to traditional finetuning [Song et al., 2024, Lee et al., 2023], evidence suggests both methods often operate through similar mechanisms—primarily modulating the model's thinking style rather than altering core knowledge [Lin et al., 2024, Zhao et al., 2025, Yang et al., 2024]; ICL's parameter-free nature, and adaptability make it a widely adopted paradigm for this purpose.

While ICL offers flexibility, it suffers from a notable performance ceiling when applied to multi-faceted tasks requiring integration of diverse information sources [He et al., 2024, Li et al., 2023b, Kirk et al., 2023]. In such scenarios, LLM agents frequently exhibit degeneration of thought, lack of diversity, or inability to follow multiple requirements [Liang et al., 2023, Huang et al., 2023, Kamoi et al., 2024, Lu et al., 2024] when using ICL. Despite increasing empirical studies on ICL's

limitations, the root causes remain under-explored. Concurrently, recent efforts to overcome the ceiling via agent-based solutions have yielded limited success, often relying on heuristics without cognitive grounding [Liu et al., 2023, Zhang et al., 2024c].

To address the first challenge—the lack of theoretical understanding behind performance ceiling—we turn to cognitive science for explanatory insight. Similar patterns of performance degradation have long been studied in cognitive science, where complex tasks involving high element interactivity often induce [Sweller, 2011, 2003]. According to Cognitive Load Theory (CLT), cognitive overload happens when working memory capacity is exceeded [Baddeley et al., 1986b]. Recent work suggests LLMs also exhibit bounded working memory with human-like failure modes under overload [Zhang et al., 2024b, Gong et al., 2024]. These shared characteristics allow us to draw an analogy that explains the observed performance degradation in LLM agents: *The performance ceiling observed in LLM agents arises when their effective cognitive load capacity is exceeded, closely mirroring the theoretical limits described by CLT.*

Building on this analogical reasoning above—that the performance ceiling observed when applying In-Context Learning (ICL) to complex tasks stems from cognitive overload—we present *CoThinker*, a multi-agent ICL architecture that directly operationalizes insights from CLT to enhance the effectiveness of ICL and improve reasoning capacity through structured cooperation among LLM agents. Specifically, *CoThinker* translates the concept of collective working memory [Kirschner et al., 2018] into a practical architecture. Just as human groups distribute cognitive demands through division of labor and shared memory structures [Wilson et al., 2004, Dunbar, 1998, Tomasello, 2009], *CoThinker* employs specialized agents for parallel thinking and coordinates their outputs via a shared memory mechanism. This collaborative architecture enables the LLM agents to offload and manage high element interactivity, thereby mitigating the cognitive overload experienced by individual agents. To demonstrate the effectiveness of leveraging CLT in this manner, we test *CoThinker* on a range of complex general problem-solving tasks and specifically fabricated high cognitive load scenarios. In sum, this paper makes the following key contributions:

- First, we are the first to explain the performance ceiling of using ICL in LLM agents by drawing a strong analogy to Cognitive Load Theory, suggesting that these limitations stem from exceeding the LLM's effective cognitive load capacity.
- Second, based on these theoretical insights, we design and introduce *CoThinker*, a novel multi-agent ICL architecture. *CoThinker* operationalizes CLT principles, through agent specialization, transactive memory, and communication moderator to mitigate cognitive overload and enhance complex cooperation.
- Third, we empirically validate *CoThinker* on complex tasks, demonstrating its ability to surpass existing multi-agent baselines. Furthermore, our analysis uncovers characteristic interaction patterns among agents, providing insights into the emergence of collective cognition within the architecture.

## 2 Related Work

### 2.1 Multi-Agent LLM Collaboration

The development of LLMs has catalyzed significant research into multi-agent systems (MAS) where LLMs function as collaborative agents, aiming to tackle more complex problems than single agents can alone [Guo et al., 2024, Wang et al., 2024a, Qian et al., 2025]. Current approaches explore various interaction structures including multi-agent debate, where agents exchange and critique ideas [Liang et al., 2023, Lu et al., 2024, Wang et al., 2024b, Du et al., 2023], iterative reflection mechanisms, enabling agents to self-correct [Shinn et al., 2023, Madaan et al., 2023, Yao et al., 2023]. Role-playing and functional specialization are also prominent, assigning distinct tasks or personas to different agents to divide labor, particularly in complex, multifaceted domains [Li et al., 2023a, Qian et al., 2023, Hong et al., 2023]. Architecturally, research investigates optimal communication topologies to enhance information flow [Li et al., 2024b], the dynamic formation and adaptation of agent networks [Liu et al., 2023, Wu et al., 2023], diversity of mental set [Liu et al., 2025b], and hierarchical structures for coordination [Zhang et al., 2024a]. However, while these systems demonstrate advancing capabilities, their designs often draw from intuition or focus on communication efficiencies, with less explicit grounding in cognitive theories that explain effective collaboration and the management of processing limitations [Pan et al., 2025]. Specifically, the systematic integration

of Cognitive Load Theory (CLT) [Sweller, 2011] remains largely underexplored in the design of LLM MAS. Our work, *CoThinker*, directly addresses this gap by operationalizing CLT to mitigate cognitive overload in LLMs and enhance collective problem-solving.

## 2.2 LLM for Human Simulation

The capacity of Large Language Models (LLMs) to exhibit human-like intelligence [Liu et al., 2025a] and emulate nuanced social behaviors [Zhou* et al., 2024] is foundational to their use as artificial agents. Research has demonstrated LLMs' ability to simulate human decision-making [Xie et al., 2024], generate believable individual and collective behaviors in social simulations [Chuang et al., 2024a], and adopt distinct personas [Chuang et al., 2024b] Critically, these parallels extend to cognitive characteristics; recent studies suggest LLMs possess bounded working memory and exhibit failure modes under cognitive overload akin to humans [Zhang et al., 2024b, Gong et al., 2024], as discussed in our introduction. Furthermore, interactions between LLM agents can mirror social psychological phenomena [Zhang et al., 2024c, Guo et al., 2024]. This confluence of human-like cognitive traits, including limitations, and social capabilities provides a strong rationale for applying principles from human cognitive science—particularly theories like Cognitive Load Theory (CLT) that address cognitive limits—to the design of more effective LLM-based collaborative systems.

# 3 Cognitive Foundations for Enhanced LLM Performance

This section establishes the theoretical basis for our approach by drawing parallels between human cognitive limitations and observed performance ceilings in LLMs. We begin (Section 3.1) by discussing analogous constraints in working memory between humans and LLMs, a concept central to Cognitive Load Theory (CLT). Building on this, we then (Section 3.2) use CLT to interpret LLM performance degradation under complex task demands. Subsequently (Section 3.3), we examine how humans overcome individual cognitive limitations by naturally forming collective cognitive systems, and finally, we posit that these principles can inform the design of a more capable LLM architecture.

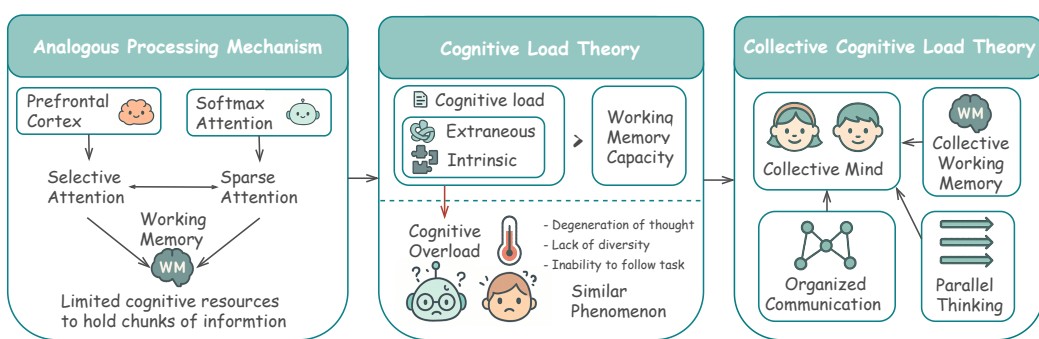

Figure 1: Analogical reasoning on how to mirror Cognitive load in human to LLM Agent to explain the performance ceiling observed when applying In-Context Learning (ICL) to LLM Agents for complex tasks, and use Cognitive Load Theory (CLT) to resolve it.

## 3.1 Working Memory Analogies

Human cognition relies fundamentally on working memory, a capacity-limited cognitive system associated with the prefrontal cortex, essential for temporarily holding and manipulating information during complex cognitive tasks such as reasoning and learning [Baddeley et al., 1986a, Cowan, 2010]. Human working memory can only hold a limited number of information chunks simultaneously, typically around 4 to 7 [Miller et al., 1956]. This system employs selective attention to filter and prioritize information [Roussy et al., 2021]. LLMs exhibit intriguing functional parallels; their core attention mechanisms perform a form of sparse, selective focus on input data [Vaswani et al., 2017]. Recent studies have begun to characterize a functional "working memory" in LLMs, identifying capacity limits and failure modes under high informational demands that echo human working memory phenomena [Zhang et al., 2024b, Gong et al., 2024]. Thus, a key analogy emerges: both humans and LLMs operate with limited cognitive resources for the concurrent processing of

information, providing a shared foundation for understanding their processing constraints. This analogy sets the stage for applying cognitive theories developed for human reasoning to interpret performance limits in LLMs (See details in Appendix).

## 3.2 Cognitive Load and Performance Limits

The finite nature of working memory is central to CLT [Sweller et al., 1998, Sweller, 2011]. CLT distinguishes between *intrinsic load*, determined by the inherent complexity and element interactivity of a task, and *extraneous load*, which can arise from how a task or its accompanying instructions are presented. When the combined load exceeds working memory capacity, *cognitive overload* ensues in humans [Paas et al., 2003, Sweller, 2011]. The provided guidance, meant to help, can paradoxically hinder performance if it contributes to exceeding cognitive capacity. LLM agents demonstrate analogous performance degradation when LLM agents are tasked with complex problems and guided by In-Context Learning (ICL). This often causes agents to fail at tasks they are capable of solving. For instance, tasks requiring extensive multi-step reasoning or the integration of numerous, potentially conflicting, constraints via ICL can lead to degeneration of thought, lack of diversity, or inability to follow multiple requirements [Liang et al., 2023, Huang et al., 2023, Kamoi et al., 2024, Lu et al., 2024] (further illustrated in Appendix). This often causes agents to fail at tasks they, in principle, are capable of solving. Drawing upon the working memory analogies and these observed patterns, we contend that such performance ceilings when applying ICL in LLMs can be understood as a manifestation of cognitive overload, where total demands surpass their effective processing capacity. To identify ways to alleviate this overload, we next examine how humans naturally overcome similar limitations through collective cognition.

## 3.3 Human Collective Intelligence

To surmount individual cognitive limitations, humans exhibit a capacity for collaborative problem-solving, leading to the emergence of a *collective intelligence* or *collective mind* that is more powerful than the sum of its individual constituents [Woolley et al., 2010, Malone et al., 2010, Shteynberg et al., 2023]. This is not simply an aggregation of independent efforts but results from sophisticated social-cognitive abilities, including shared intentionality, theory of mind, and nuanced communication for establishing common ground [Tomasello et al., 2005, Frith and Frith, 2005]. Such collective entities effectively expand cognitive resources by distributing processing. Key aspects include the formation of a *collective working memory*, often through Transactive Memory Systems where knowledge and responsibilities are shared [Wegner, 1987, Kirschner et al., 2018] and individuals have meta-knowledge about who knows what [Hollingshead, 2001] so that they can rely on each other for information sharing and retrieval [Hollingshead and Brandon, 2003]; the engagement in *parallel thinking* through a division of cognitive labor, which reduces the intrinsic load on each individual [Dunbar, 2003]; and the use of *organized communication* to integrate diverse information and maintain a shared attentional focus [Hutchins, 1995]. These spontaneously formed group structures allow humans to manage complexities that would overwhelm an individual, demonstrating a natural solution to cognitive overload.

Inspired by these human collective cognitive strategies and human-LLM cognitive similarity discussed above, the subsequent section introduces *CoThinker*, a multi-agent ICL architecture designed to operationalize these principles to overcome LLM performance ceilings whe using ICL.

## 4 CoThinker

*CoThinker* is a multi-agent ICL architecture designed to enhance collaborative problem-solving by systematically managing cognitive load. Simply aggregating outputs from LLM agents often proves insufficient for complex tasks, as naive collaboration can introduce significant transactional costs—the cognitive effort required to coordinate, communicate, and integrate—without a corresponding increase in solution quality [Pan et al., 2025]. As Cognitive Load Theory (CLT) suggests, these transactional costs can quickly lead to extraneous cognitive overload, negating the benefits of parallel thinking [Kirschner et al., 2009, 2018]. To overcome these challenges within the ICL paradigm, *CoThinker* operationalizes the principles of human collective intelligence discussed in Section 3, aiming to create a "collective mind" that distributes cognitive load. We leverage the insights from CLT to design an architecture that mirrors how human groups effectively solve complex problems.

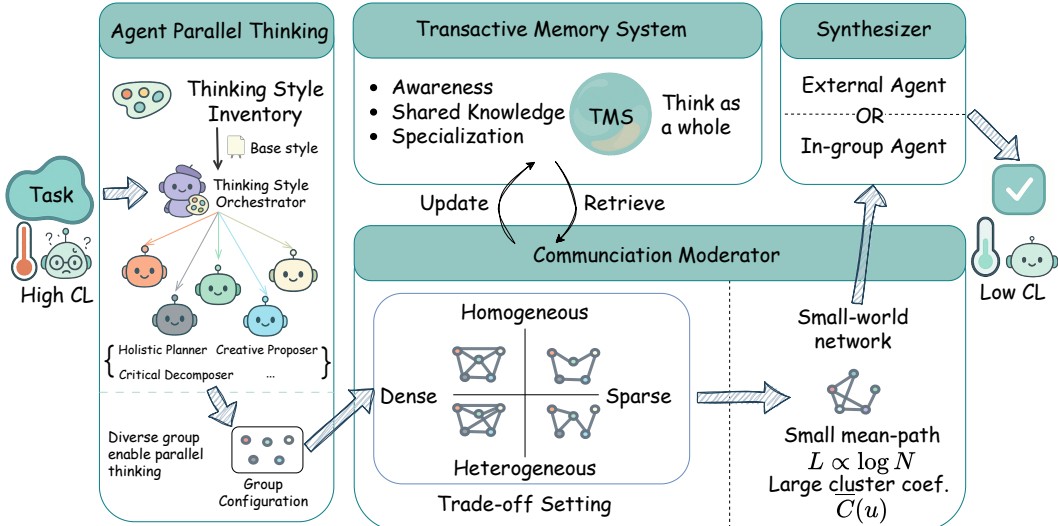

Figure 2: The *CoThinker* Architecture. A high Cognitive Load (CL) task is initially processed by diverse agents via Agent Parallel Thinking. The Transactive Memory System (TMS) facilitates shared understanding by updating and allowing retrieval of collective knowledge. The Communication Moderator manages inter-agent information flow, leveraging a trade-off to form a cognitive small-world network, which then feeds into the Synthesizer to produce a final solution, resulting in a lower effective CL for the overall system.

To operationalize these insights, the *CoThinker* architecture (Figure 2) comprises four main modules: Agent Parallel Thinking (Section 4.1), Transactive Memory System (Section 4.2), Communication Moderator (Section 4.3), and Synthesizer (Section 4.4). Each module is directly guided by CLT principles to emulate aspects of the human collective mind. Agent Parallel Thinking fosters initial cognitive diversity, potentially splitting the intrinsic load of the task. The Transactive Memory System boosts inter-agent understanding and tracks consensus, reducing cognitive load from redundant processing. The Communication Moderator balances intrinsic and extraneous loads by structuring information exchange. Finally, the Synthesizer integrates refined collective insights. Let $\mathcal{A} = \{A_1, \ldots, A_M\}$ be the set of $M$ agents. Let $T_{\max}$ be the total number of generation rounds. Agent $A_i$'s output at the end of round $t$ is denoted $x_i^{(t)}$.

## 4.1 Agent Parallel Thinking

This module promotes a *division of cognitive labor* and *parallel thinking* by assigning diverse thinking styles. Unlike assigning pre-defined roles, which require domain-specific foresight and impose extraneous cognitive load from role adherence, *CoThinker* uses an adaptive approach. A Thinking Style Orchestrator generates a task-specific style $\phi_i$ for each agent $A_i$ based on a general base thinking style inventory $\psi$ [Sternberg, 1997] and the task $D$:

$$\{\phi_i\}_{i=1}^M = \text{Orch}(D, \psi) \tag{1}$$

This yields diverse thinking styles $\{\phi_i\}_{i=1}^M$, employed in subsequent stages. Further details on the prompting strategy for style generation and thinking style inventory are in the Appendix.

## 4.2 Transactive Memory System (TMS)

Human groups effectively manage complex information by developing Transactive Memory Systems (TMS), which involve a shared understanding of who knows what, how to access information held by others, and a collective agreement on the information itself [Wegner, 1987, Hollingshead, 2001]. This distributed cognitive system allows individuals to specialize and rely on others, reducing individual cognitive load and enhancing group problem-solving [Lewis, 2003]. To emulate these benefits and foster a *collective working memory* in *CoThinker*, we implement a structured mechanism for maintaining and accessing shared knowledge. At each round $t$, an evolving representation of the

group's collective knowledge, denoted $\mu^{(t)}$, is updated based on contributions from all agents:

$$\mu^{(t+1)} = \text{UpdateMem}(\mu^{(t)}, \{x_j^{(t)}\}_{j=1}^M) \tag{2}$$

This aims to enhance shared awareness and efficient integration of distributed knowledge. The specific prompt-based emulation of TMS components is detailed in the Appendix.

### 4.3 Communication Moderator

Effective inter-agent communication is crucial, yet it incurs transactional costs—the cognitive effort for message processing and integration—which can impose extraneous cognitive load, a key concern in Collaborative Cognitive Load Theory [Kirschner et al., 2009, 2018]. To mitigate these costs, the Communication Moderator structures information exchange by selecting $N$ reference messages $\mathcal{P}_i^{(t-1)}$ for each agent $A_i$. This process navigates the critical trade-offs between **Network Density vs. Sparsity** (high exposure and cost vs. low cost and potential information loss) and **Information Homogeneity vs. Heterogeneity**. The latter involves balancing the ease of integrating cognitively similar inputs (low extraneous load but risk of echo chambers [Runkel, 1956]) against the benefits of diverse perspectives for distributing intrinsic load [Aral and Van Alstyne, 2011].

**Communication Topology and Algorithm:** The selection of references defines a directed communication graph $G^{(t-1)} = (\mathcal{A}, E^{(t-1)})$ for each round, where an edge $(A_u, A_v) \in E^{(t-1)}$ exists if agent $A_v$ receives a message from agent $A_u$ generated in round $t-1$. Motivated by how small-world networks efficiently balance local clustering with global connectivity [Watts and Strogatz, 1998], our moderator employs the following algorithm to construct this graph:

a. **Set Fixed In-Degree** ($N$)**:** Each agent $A_i$ (node $A_v$) has an in-degree of $N$, capping its processing load and respecting LLM working memory [Zhang et al., 2024b, Gong et al., 2024].

b. **Define Cognitive Distance between Agent Outputs:** The cognitive distance $d(x_u^{(t-1)}, x_v^{(t-1)}) = 1 - \text{sim}(x_u^{(t-1)}, x_v^{(t-1)})$ is based on the semantic similarity of previous outputs.

c. **Connection Establishment via Probabilistic Rewiring** ($\beta$)**:** For each agent $A_i$, its $N$ incoming edges (references $\mathcal{P}_i^{(t-1)}$) are established by primarily choosing messages from cognitively similar peers (low distance), but with a probability $\beta$, "rewiring" some connections to randomly chosen, diverse peers.

**Resulting Network Properties and Cognitive Balance:** This rewiring process fosters dynamic communication networks $G^{(t-1)}$ with small-world properties. Such networks exhibit high local clustering (facilitating efficient refinement of similar ideas, reducing extraneous load locally) and short average path lengths (enabling rapid global propagation of diverse insights, aiding intrinsic load distribution). This structure offers a principled balance between focused collaboration and broad information access, managing cognitive load more effectively than purely random or regular lattice networks. Further details are in the Appendix.

### 4.4 Synthesizer

The Synthesizer consolidates information into a final solution after $T_{max}$ rounds. It can be an External Agent (dedicated LLM) or an In-group Agent (team member) [Lu et al., 2024, Shinn et al., 2023]. This draws from Collaborative Cognitive Load Theory [Kirschner et al., 2018] and Observational Learning [Bandura and Walters, 1977] (See details in Appendix)

#### *CoThinker* Process Flow

The process for task $D$ with $M$ agents over $T$ rounds:

*Initialization:*

$$\{\phi_i\}_{i=1}^M = \text{Orch}(D, \psi_i), \quad x_i^{(0)} = \text{Agent}(D, \phi_i), \quad \mu^{(0)} = \text{UpdateMem}(\{x_i^{(0)}\}_{i=1}^M) \tag{3}$$

*Iterative Refinement:* For each agent $A_i$ and round $t$:

$$\mathcal{P}_i^{(t)} = \text{SelectRefs}(\{x_k^{(t)}\}_{k \in \mathcal{A} \setminus \{A_i\}}, x_i^{(t)}, N, \beta) \tag{4}$$

$$x_i^{(t+1)} = \text{Agent}(D, \phi_i, \mu^{(t)}, x_i^{(t)}, \mathcal{P}_i^{(t)}) \tag{5}$$

$$\mu^{(t+1)} = \text{UpdateMem}(\mu^{(t)}, \{x_i^{(t+1)}\}_{i=1}^M) \tag{6}$$

247 *Final Synthesis:*

$$y_{\text{final}} = \text{Synth}\big(\{x_i^{(T-1)}\}_{i=1}^M, \mu^{(T-1)}, D\big) \qquad (7)$$

# 248 5 Experiments and Results

249 This section details our experimental methodology and presents the empirical evaluation of *CoThinker*.
250 We first outline the experimental setup, including the base LLMs, benchmarks, and baselines. We
251 then present the main results on LiveBench and CommonGen-Hard, followed by ablation studies and
252 a discussion of our findings through the lens of Cognitive Load Theory (CLT).

## 253 5.1 Experimental Setup

254 **Models and Configuration.** We use three Gemini models [Team et al., 2024] with varying capaci-
255 ties: `gemini-1.5-flash-8b` (lightweight), `gemini-1.5-flash` (mid-tier), and `gemini-1.5-pro`
256 (high-capacity). All models run with the initial generation temperature set to 0.25 to encourage
257 diverse outputs. In multi-agent settings, subsequent rounds use temperature 0.0 and a frequency
258 penalty of 0.5 to reduce repetition. By default, multi-agent methods use $M{=}6$ agents interacting over
259 $T{=}3$ rounds. For *CoThinker*, we set $N{=}3$ references and exploration parameter $\beta{=}0.3$.

260 **Evaluation Benchmarks.** We evaluate on two challenging benchmarks: (1) **LiveBench** [White
261 et al., 2025], a recent diverse suite drawing from Big-Bench Hard [Suzgun et al., 2023], AMPS
262 [Hendrycks et al., 2021], and IFEval [Zhou et al., 2023], covering domains such as mathematics,
263 coding, language, instruction following, and data analysis; and (2) **CommonGen-Hard** [Madaan
264 et al., 2023], a cognitively demanding variant of CommonGen [Lin et al., 2020], which evaluates
265 multi-sentence generation under high element interactivity. We adopt a 10-dimensional metric for
266 CommonGen-Hard evaluation [Li et al., 2018]. See full details in the Appendix.

267 **Baselines.** We compare *CoThinker* with both single-agent and multi-agent approaches. *(i) Single
268 Agent (IO)* is a standard mode of prompting. *(ii) Single Agent (CoT)* incorporates Chain-of-Thought
269 prompting [Wei et al., 2022]. *(iii) Single Agent (Self-Refine)* uses iterative self-critique and revision
270 [Madaan et al., 2023]. *(iv) Multi-Agent Debate (MAD):* employs interactive agent discussion with
271 consensus formation [Du et al., 2023, Liang et al., 2023]. *(v) Diverse MAD (DMAD):* introduces
272 heterogeneous prompting to avoid fixed mental sets [Liu et al., 2025b]. See details in the Appendix.

## 273 5.2 Main Results on LiveBench

274 Table (1) presents the performance of *CoThinker* and baseline methods across the LiveBench suit
275 for `gemini-1.5-flash-8b`, `gemini-1.5-flash`, and `gemini-1.5-pro`. Scores are reported as
276 relative improvements over the respective `gemini-8b-flash`'s IO (Standard Prompt) baseline. An
277 average score is calculated as the arithmetic mean of these relative scores across the main LiveBench
278 categories (Math, Reasoning, Instruction, Data, Language).

| Method | gemini-1.5-flash-8b | | | | | | gemini-1.5-flash | | | | | | gemini-1.5-pro | | | | | |
|---|---|---|---|---|---|---|---|---|---|---|---|---|---|---|---|---|---|---|
| | Math | Data | Reas. | Lang. | Instr. | Avg. | Math | Data | Reas. | Lang. | Instr. | Avg. | Math | Data | Reas. | Lang. | Instr. | Avg. |
| IO | 1.00 | 1.00 | 1.00 | 1.00 | 1.00 | 1.00 | 1.47 | 2.03 | 1.63 | 1.41 | **1.10** | 1.53 | 2.00 | 2.92 | 1.87 | 1.43 | **1.03** | 1.85 |
| CoT | 1.04 | 0.90 | 1.11 | **1.09** | **1.02** | 1.03 | 1.47 | 2.07 | 1.74 | 1.30 | **1.10** | 1.54 | 1.86 | 2.72 | 1.82 | 1.54 | 1.02 | 1.79 |
| SR | 0.92 | 0.34 | 0.80 | 0.89 | 0.81 | 0.75 | 1.45 | 0.90 | 1.55 | 1.06 | 0.87 | 1.17 | 1.93 | 1.33 | 1.80 | 1.22 | 0.72 | 1.40 |
| MAD | **1.13** | 0.58 | 1.21 | 1.03 | 0.87 | 0.97 | 1.51 | 1.46 | 1.92 | 1.46 | 1.01 | 1.47 | 2.29 | 3.15 | 1.78 | 1.58 | 0.77 | 1.92 |
| DMAD | **1.13** | 0.64 | 0.85 | 1.02 | 0.89 | 0.91 | 1.49 | **2.51** | 1.94 | 1.44 | 1.06 | 1.69 | 2.31 | 3.32 | 1.88 | 1.74 | 1.02 | 2.05 |
| *CoThinker* | 1.11 | **1.32** | **1.22** | 0.98 | 0.80 | **1.07** | **1.57** | 2.44 | **1.97** | **1.52** | 0.99 | **1.70** | **2.40** | **3.39** | **1.95** | **1.76** | 0.95 | **2.09** |

Table 1: LiveBench[White et al., 2025] performance with all scores normalized by
`gemini-1.5-flash-8b-io` baseline. The abbreviations corresponded to Math, Data Analysis,
Reasoning Language, and Instruction Following

**Analysis of LiveBench Results.** *CoThinker* consistently achieves strong average performance across
280 all base model families, with particularly notable gains in complex categories like Data Analysis,
281 Reasoning, and often Math, but low performance on Instruction Following. We posit this perfor-
282 mance pattern reflects two distinct task categories: those with high intrinsic cognitive load and
283 those with low intrinsic load. The former, characterized by tasks like Data Analysis and Reason-
284 ing, demonstrates a clear scaling in baseline performance as model capability increases (e.g., from

gemini-1.5-flash-8b to gemini-1.5-pro), indicating that greater raw cognitive power inherently improves outcomes. For these high-load tasks, *CoThinker* excels by effectively decomposing complex problems and orchestrating collaborative agent contributions, therefore, splitting the intrinsic load to enhance performance.

Conversely, tasks with low intrinsic load, such as instruction following (Instr.), show minimal or inconsistent performance gains when moving from weaker to stronger base models; for instance, the gemini-1.5-pro IO baseline on Instruction Following does not substantially outperform that of gemini-1.5-flash-8b. This suggests the primary bottleneck is not cognitive load. In such scenarios, the added communication overhead inherent in *CoThinker* can outweigh the benefits of collaboration. For tasks demanding straightforward adherence rather than sophisticated reasoning, this introduced more extraneous cognitive load, explaining why *CoThinker* may not show an advantage or can even underperform on these low-load, execution-focused tasks.

## 5.3 Main Results on CommonGen-Hard

In CommonGen-Hard, which emphasizes managing high element interactivity, *CoThinker* demonstrates notable performance improvements. Figure 3 presents these results, with Figure 3a illustrating its balanced strengths across evaluation dimensions and Figure 3b showing performance trends over interaction rounds.

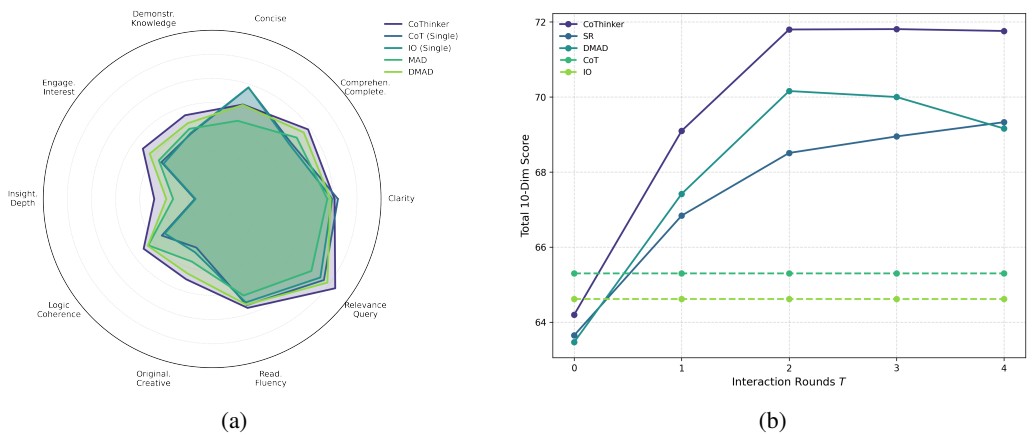

|  (a)  |  (b)  |

Figure 3: *CoThinker* performance on CommonGen-Hard using gemini-1.5-flash. (a) The radar plot illustrates a multi-dimensional performance profile, where *CoThinker* typically shows well-rounded and superior strengths compared to baselines. (b) The rounds plot depicts the total score trend across interaction rounds ($T$), often indicating an optimal number of rounds before performance plateaus or declines.

**Analysis of CommonGen-Hard Results.**
*CoThinker* demonstrates strong overall performance on CommonGen-Hard (Figure 3), effectively managing its high element interactivity. The multi-dimensional profile (Figure 3a) typically shows *CoThinker* excelling in key areas like coherence and concept integration, albeit with occasional trade-offs in aspects such as conciseness. This aligns with Cognitive Load Theory (CLT); the high intrinsic load of the task is managed by *CoThinker*'s distributed processing and transactive memory. Notably, its performance trajectory versus interaction rounds (Figure 3b) highlights a key advantage: *CoThinker* achieves sustained constructive refinement over several rounds, effectively managing cognitive load. This contrasts with the multi-agent baseline where performance degrades due to rapidly accumulating extraneous load from inefficient coordination or information overload. *CoThinker*'s architecture appears more adept at balancing these loads, delaying diminishing returns.

## 5.4 Ablation Studies on LiveBench Subsets

Ablation studies were conducted on gemini-1.5-flash-8b using averaged scores from selected LiveBench subtask categories (Math, Reasoning, Data Analysis, and Instruction). These studies investigated the impact of *CoThinker*'s reference set size ($N$), exploration rate ($\beta$), and the number of agents ($M$). Unless otherwise specified, default parameters were $T = 3$. For $N$ ablation,

318 $M = 6, \beta = 0.3$. For $\beta$ ablation, $N = 2, M = 6$. For $M$ ablation, $N = 3, \beta = 0.3$. All scores are
319 normalized by the I/O baseline performance for each subtask before category averaging.

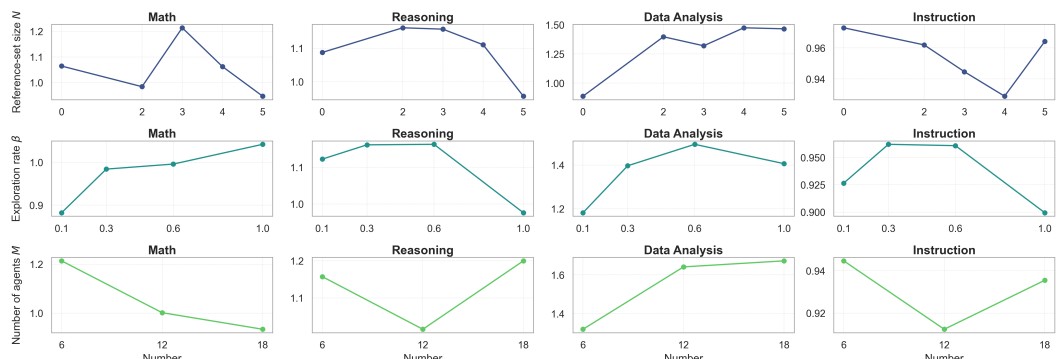

Figure 4: Ablation studies on *CoThinker* parameters $(N, \beta, M)$ using `gemini-1.5-flash-8b`. Performance is shown for four LiveBench task categories (Math, Reasoning, Data Analysis, Instruction), normalized by IO baseline performance (1.0). **Top row**: Effect of Reference Set Size ($N$), varying $N \in \{0, 2, 3, 4, 5\}$ with $M = 6, \beta = 0.3, T = 3$. **Middle row**: Effect of Exploration Rate ($\beta$), varying $\beta \in \{0.1, 0.3, 0.6, 1.0\}$ with $N = 2, M = 6, T = 3$. **Bottom row**: Effect of Number of Agents ($M$), varying $M \in \{6, 12, 18\}$ with $N = 3, \beta = 0.3, T = 3$. Optimal parameter settings are task-dependent, indicating varying sensitivities to peer input diversity and information overload.

320 **Analysis of Ablation Studies.**
321 Figure 4 demonstrates *CoThinker*'s hyperparameter sensitivity, offering insights into cognitive load
322 management as theorized in Section 3. The reference set size ($N$, top row) directly impacts extraneous
323 cognitive load. An optimal $N$ (e.g., 2-3) balances diverse peer input against cognitive overload,
324 respecting LLM working memory limits. Too few references limit collaboration; too many overwhelm.
325 The exploration rate ($\beta$, middle row) governs the trade-off between exploiting similar ideas (low $\beta$,
326 lower extraneous load for integration) and exploring diverse ones (high $\beta$, high extraneous load). Task-
327 dependent optima, like higher $\beta$ for Reasoning, reflect this balance, managed by the Communication
328 Moderator's cognitive small-world network. The number of agents ($M$, bottom row) shows that while
329 more agents can distribute intrinsic load, increasing $M$ also elevates transactional (extraneous) load
330 from coordination. Non-monotonic performance indicates that beyond a point, these transactional
331 costs negate the benefits of parallelism, aligning with CLT's predictions for group overload. These
332 findings affirm that *CoThinker*'s parameters are crucial for managing cognitive load, enabling the
333 emergence of an effective "collective mind" by mitigating overload.

334 # 6   Conclusion

335 This work addresses the performance limitations of LLMs on complex tasks, particularly when
336 employing In-Context Learning (ICL), by drawing an analogy to Cognitive Load Theory (CLT). We
337 posit that observed performance ceilings arise from exceeding an LLM's effective cognitive load
338 capacity when processing intricate task details and extensive in-context guidance. We introduced
339 *CoThinker*, a multi-agent architecture that operationalizes CLT principles. Through agent specializa-
340 tion, a transactive memory system, and moderated communication, *CoThinker* mitigates overload
341 and enhances collaborative problem-solving, especially for tasks that challenge single agents using
342 ICL. Empirical evaluations on benchmarks like LiveBench and CommonGen-Hard demonstrated
343 *CoThinker*'s superior performance over existing baselines on high-load tasks. Analyses validated
344 *CoThinker*'s effective management of cognitive load, fostering a more robust "collective mind." By
345 grounding multi-agent LLM design in CLT, this research offers a principled path towards overcoming
346 performance bottlenecks encountered when applying ICL to demanding problems, contributing to
347 more powerful collaborative AI systems through the lens of cognitive science.

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

**Appendix**



# A  Cognitive Foundations: Elaborations

## A.1  Human Working Memory and Attentional Control

Human working memory (WM) is a core cognitive faculty for actively holding and manipulating a limited amount of information relevant to ongoing tasks, operating through attentional mechanisms that select and maintain internal representations, often associated with sustained neural activity in regions like the prefrontal cortex [Baddeley et al., 1986a, Cowan, 2010, Postle, 2006]. Given that Large Language Models exhibit emergent sparse attention—where specific attention heads specialize in processing distinct patterns rather than diffusely attending to all input tokens [Vaswani et al., 2017, Voita et al., 2019]—it prompts an intriguing question: does this selective information processing within a finite context window imply the existence of a functional analogue to human WM in LLMs? This emergent selectivity, where not all information in the context is equally weighted or actively processed at any given step, forms a crucial part of the analogy we draw to understand potential capacity limitations and cognitive load phenomena in these models, particularly when handling tasks with high element interactivity through In-Context Learning.

## A.2  Using Cognitive Load Theory to Explain Phenomena in LLM Performance

Cognitive Load Theory (CLT) offers a valuable lens to interpret puzzling LLM performance issues, positing that LLMs, like humans, have finite processing capacity. Exceeding this capacity leads to performance degradation. This section concisely analyzes several such cases through CLT.

1. **Degradation of Thought in Self-Reflection:** Liang et al. [2023] found LLMs may rigidly stick to incorrect initial answers during self-reflection, failing to correct meaningfully.
   - *CLT Explanation:* Self-reflection (holding problem, solution, critique, and revision process concurrently) is highly demanding. If initial analysis already consumes most capacity, the LLM may lack resources for genuine re-evaluation, defaulting to superficial agreement due to cognitive overload.
2. **Performance Degradation with More In-Context Examples (Many-Shot ICL):** Agarwal et al. [2024] noted LLM performance can degrade with more in-context examples, especially on complex tasks (e.g., MATH).
   - *CLT Explanation:* While few examples scaffold, excessive examples increase total cognitive load beyond capacity. The LLM struggles to synthesize all information, akin to CLT's "redundancy effect" where too much information, even relevant, overwhelms working memory.
3. **Performance Degradation Despite Increasing "Confidence" (NLL Trends):** Agarwal et al. [2024] also found that performance degradation in many-shot ICL wasn't always explained by NLL (confidence) trends; NLL could improve as performance worsened.
   - *CLT Explanation:* Under cognitive overload, LLMs (like humans) may resort to heuristics. Overwhelmed by many examples, an LLM might latch onto superficial patterns, yielding outputs that are stylistically plausible (good NLL) but incorrect. This "overconfidence" in a flawed heuristic stems from an inability to allocate resources for deeper reasoning.
4. **Reduced Diversity after RLHF for Instruction Following:** Kirk et al. [2023] and others observed that RLHF, while improving instruction following, can reduce output diversity.
   - *CLT Explanation:* Intense RLHF training on narrow preferences imposes a high "germane load" for conformance. To manage this, and the extraneous load of deviating from rewarded paths, the model may operate in a constrained output space, reducing the cognitive effort of exploring diverse (potentially unrewarded) responses. The "cost" of diversity becomes too high.

These instances suggest CLT is a powerful analogical framework for understanding LLM limitations under demanding informational or processing conditions.

## B  CoThinker Architecture: Implementation and Prompting

### B.1  Prompt Architecture for Agent Parallel Thinking

The Agent Parallel Thinking module in CoThinker aims to foster a beneficial division of cognitive labor by assigning diverse thinking styles to agents. This approach is grounded in theories of thinking styles, such as Sternberg's Theory of Mental Self-Government [Sternberg, 1997], which posits that styles are preferred ways of using one's abilities, not abilities themselves. This distinction is crucial: CoThinker leverages thinking styles as preferential orientations for LLM agents, assuming the base model possesses a broad set of underlying capabilities. The assigned style guides how these capabilities are applied to the task, rather than attempting to imbue a new, fixed skill or enforce a rigid behavioral script as a predefined "role" might. This aligns with findings that In-Context Learning often modulates an LLM's thinking style rather than altering its core knowledge [Lin et al., 2024, Zhao et al., 2025].

Adherence to a flexible thinking style is hypothesized to impose less extraneous cognitive load on an LLM agent compared to maintaining a complex, predefined role persona. This allows more of the agent's cognitive resources to be dedicated to the primary task. Furthermore, while core thinking styles are often seen as relatively stable, they are also understood to be somewhat malleable and can be adapted to specific task demands [Sternberg, 1997]. CoThinker operationalizes this adaptability through a two-stage prompting strategy:

**1. Style Orchestration ($\mathrm{Orch}$ function):** The Thinking Style Orchestrator (itself an LLM) is provided with the overall task description $D$ and a Thinking Style Inventory. This inventory consists of base thinking styles derived from Sternberg's theory, encompassing dimensions such as Functions (Legislative, Executive, Judicial), Forms (e.g., Monarchic, Hierarchic), Levels (Global, Local), Scope (Internal, External), and Leanings (Liberal, Conservative). The Orchestrator's objective is to generate a diverse yet task-relevant set of $M$ specific thinking styles $\{\phi_1, \ldots, \phi_M\}$, one for each agent $A_i$. For each agent, the Orchestrator takes one or a combination of Sternberg's dimensions as a base style $\psi_i$ and adapts it to the given task $D$. The Orchestrator is guided to ensure the resulting set of styles $\{\phi_i\}$ promotes varied perspectives on the problem, reflecting the value of different styles for different task facets.

An example prompt for the Orchestrator, given a base combination from Sternberg (e.g., $\psi_i =$ "Legislative-Global style"):

```
Given the primary task: "{Task D}"
And the base thinking style profile (from Sternberg's Theory of
Mental Self-Government): "{Base Style profile psi_i, e.g.,
Legislative function with a Global level preference}"

Generate a concise (1-2 sentences) task-specific adaptation
of this thinking style profile that would be most beneficial
for an agent contributing to this primary task. The agent
should focus its reasoning and output according to this
adapted style.
Task-Specific Style for an agent:
```

This process results in $M$ distinct, task-contextualized thinking styles $\{\phi_1, \ldots, \phi_M\}$. By dynamically adapting general styles to the specific task, CoThinker aims to harness the benefits of stylistic diversity while mitigating risks such as pigeonholing or oversimplification associated with static style assignments.

**2. Agent Instruction ($\mathrm{Agent}$ function - style incorporation):** Each agent $A_i$ then receives its specific thinking style $\phi_i$ as part of its instruction prompt, guiding its approach throughout the problem-solving process. An excerpt of an agent's prompt showing style incorporation:

```
You are Agent {num}. Your assigned thinking style for this
task is: "{Style phi_i generated by Orchestrator}".
The overall task is: "{Task D}".
[Other contextual information, e.g., from TMS mu^(t),
references P_i^(t-1), own previous thought x_i^(t-1)]
```

```
Keeping your assigned thinking style in mind, please provide
your thoughts/solution:
```

This method encourages agents to approach the problem from varied cognitive angles, promoting comprehensive exploration of the solution space and distributing the intrinsic cognitive load of the task, without the cognitive burden of strict role-playing.

**B.2  Prompt Architecture for Transactive Memory System (TMS) Emulation**

As introduced in Section 4.2, CoThinker incorporates a mechanism to emulate a human Transactive Memory System (TMS). A TMS is a collective cognitive resource developed by groups, encompassing a shared understanding of who knows what (metamemory or expertise directory), how to access and integrate this distributed knowledge, and a level of trust in the information provided by different members [Wegner, 1987, Hollingshead, 2001, Lewis, 2003]. Effective TMS functioning involves processes of knowledge *encoding* (assigning information to members or recognizing expertise), *storage* (individuals retaining specialized knowledge), and *retrieval* (accessing and using the distributed knowledge), facilitated by member *specialization*, perceived *credibility*, and inter-agent *coordination* [Yoo and Kanawattanachai, 2001]. This systematic division and integration of cognitive labor allows groups to handle more complex information and solve problems more effectively than individuals or less coordinated groups.

CoThinker's emulation of TMS centers on the generation and presentation of the collective memory state, $\mu^{(t)}$, at each round $t$. This is not merely an aggregation of past messages but a structured synthesis designed to reflect key TMS components. Specifically, an auxiliary LLM agent (the "TMS Manager") is tasked with populating a predefined "TMS Template" based on all agent outputs $\{x_j^{(t-1)}\}_{j=1}^M$ from the previous round and the existing memory state $\mu^{(t-1)}$, to produce the updated $\mu^{(t)}$. This template explicitly guides the TMS Manager to synthesize information reflecting:

1. **Expertise Directory ("Who Knows What"):** The template prompts the TMS Manager to list the key contributions from each agent $A_j$ in the previous round, often implicitly linking these contributions back to their assigned thinking style $\phi_j$ or emergent problem-solving role. For example, $\mu^{(t)}$ might state: *"Agent A (Analytical Thinker) identified three inconsistencies in the data, while Agent B (Creative Ideator) proposed two novel solutions based on X."* This helps all agents maintain an updated awareness of which peer is focusing on, or has provided significant input regarding, specific facets of the task. This corresponds to the *encoding* of expertise and facilitates targeted *retrieval* cues.

2. **Shared Knowledge Store (Consensus and Artifacts):** The template requires the TMS Manager to identify and articulate points of emerging consensus, established facts, or partial solutions that the group has collectively built. For instance: *"Consensus: The primary bottleneck is resource allocation. Established: The budget cannot exceed Y."* This component of $\mu^{(t)}$ serves as the repository of *stored*, validated collective knowledge, reducing the need for agents to re-derive information and providing a foundation for subsequent reasoning.

3. **Differential Insights and Unresolved Issues (Focus for Coordination):** A crucial part of the TMS template prompts the TMS Manager to highlight discrepancies between agent outputs, unresolved questions, conflicting perspectives, or aspects of the problem that remain unaddressed. Example: *"Divergence: Agent C suggests strategy Alpha, while Agent D advocates for Beta. Unresolved: The feasibility of implementing X within the given timeframe."* This explicitly flags areas requiring further discussion, debate, or focused problem-solving in the next round, thereby guiding inter-agent *coordination* and ensuring that cognitive effort is directed towards the most critical, unresolved aspects of the task assigned to most relavent agents.

The structure of $\mu^{(t)}$, as generated by this templated process, is then presented to each agent $A_i$ at the beginning of round $t$ as part of its input prompt. An excerpt illustrating this presentation is:

```
[Agent's assigned thinking style: {Style_phi_i}]
[Overall Task: {Task_D}]

Collective Summary from Previous Round (reflecting shared understanding mu^(t)):
"{Text of mu^(t) generated by the TMS Manager using the TMS Template}"
```

```
755
756   Your Previous Output (x_i^(t-1)):
757   "{Text of x_i^(t-1)}"
758
759   Reference Outputs from Peers (P_i^(t-1)):
760   Reference 1 (from Agent A_k): "{Text of x_k^(t-1)}"
761   Reference 2 (from Agent A_l): "{Text of x_l^(t-1)}"
762   ...
763
764   Based on all the above, and keeping your thinking style in mind,
765   provide your refined thoughts/contribution for the current round:
```

This deliberate structuring of $\mu^{(t)}$ to reflect an expertise directory, a shared knowledge store, and a pointer to unresolved issues distinguishes CoThinker's approach from simple multi-agent cooperation or discussion. While basic cooperation might involve information sharing, it often lacks the systematic assignment of knowledge domains, explicit tracking of expertise, and focused mechanisms for integrating specialized insights that a TMS provides. CoThinker's TMS emulation aims to create a more efficient and powerful "group mind" by embedding these principles directly into the information environment of the agents, thereby reducing redundant effort and enhancing the quality of collective problem-solving.

### B.3 Communication Moderator: Cultivating an Efficient Network via Strong and Weak Ties

The Communication Moderator in *CoThinker* (Section 4.3) strategically structures inter-agent communication by implicitly leveraging principles from social and complex network theories. This design fosters a network optimized for managing cognitive load and enhancing collective intelligence.

**Local Cohesion via Strong Cognitive Ties and High Clustering**   The primary reference selection mechanism (with probability $1 - \beta$) connects agent $A_i$ to peers whose prior outputs $x_k^{(t-1)}$ are most cognitively similar to $A_i$'s own $x_i^{(t-1)}$. This promotes the formation of local clusters where agents process highly related information. From a social network perspective, these connections are analogous to **strong ties** [Granovetter, 1983], fostering cohesive subgroups. In network science, this behavior inherently leads to a high **local clustering coefficient**, indicating dense intra-group connectivity.

- **Rationale:** Such local clustering facilitates focused refinement of shared ideas and reduces the extraneous cognitive load associated with integrating highly similar information.

**Global Integration via Weak Cognitive Ties and Small-World Properties**   Exclusive reliance on strong ties (i.e., $\beta = 0$) could lead to network fragmentation, where clusters become isolated "echo chambers." This corresponds to a lack of "bridging capital" across **structural holes** in social network theory [Burt, 2004], and a long **average path length** in network science, hindering the global distribution of diverse insights and the effective management of overall intrinsic cognitive load.

The probabilistic "rewiring" mechanism (with probability $\beta$) counteracts this by compelling agents to also reference randomly chosen peers, irrespective of immediate cognitive similarity.

- **Mechanism and Analogy:** These random connections function as **weak ties** [Granovetter, 1983], which are crucial for bridging disparate network segments and transmitting novel information.
- **Network Outcome:** Introducing such weak ties into a highly clustered network is a hallmark of **small-world networks** [Watts and Strogatz, 1998]. These networks advantageously combine high local clustering with short global average path lengths.
- **Rationale:** In *CoThinker*, these $\beta$-driven connections ensure efficient propagation of diverse perspectives across cognitive clusters. This shortens the information path length, promotes the synthesis of varied knowledge, helps distribute the intrinsic cognitive load of the overall task, and prevents premature convergence.

In essence, the Communication Moderator dynamically cultivates a network with small-world characteristics. By balancing the formation of strong-tie local clusters for specialized processing with weak-tie bridges for global integration, it supports both deep, focused collaboration and the broad synthesis of diverse insights, crucial for effective collective problem-solving.

### B.4 Synthesizer Module: Consolidation and Cognitive Grounding

The Synthesizer module (Section 4.4) consolidates outputs from all agents ($\{x_i^{(T-1)}\}_{i=1}^{M}$) and the final Transactive Memory System state ($\mu^{(T-1)}$) into a single solution for the task $D$. The design choice for the Synthesizer can vary, with different cognitive implications:

1. **External Agent Synthesizer (Observational Learning):** This involves a dedicated LLM instance, distinct from the collaborating agents, to produce the final output. This agent receives all final individual perspectives and the collective memory summary.
   - *Cognitive Analogy:* This setup mirrors **Observational Learning** [Bandura and Walters, 1977]. The External Synthesizer observes the diverse problem-solving behaviors and refined outputs of the specialist agents. By analyzing these varied "models" of thought and their collective synthesis ($\mu^{(T-1)}$), it can construct a comprehensive solution, potentially integrating insights in a novel way without having been part of the iterative load distribution.
2. **In-group Agent Synthesizer (Collaborative Leading/Shared Regulation):** One of the existing collaborating agents (e.g., an agent identified as a leader or one with a consistently high-quality output, or a randomly chosen one) can be tasked with synthesizing the final solution. This agent uses its own understanding, the collective memory $\mu^{(T-1)}$, and the final outputs of its peers. align
   - *Cognitive Analogy:* This aligns with principles from **Collaborative Cognitive Load Theory (CCLT)** [Kirschner et al., 2018], specifically aspects of shared regulation and distributed leadership. The synthesizing agent, having participated in the collaborative process, leverages its deep contextual understanding and the established collective working memory ($\mu^{(T-1)}$) to guide the final integration. Its synthesis is an act of "collaborative leading" by taking responsibility for the final product based on the group's efforts.

**Sample Prompt for an External Agent Synthesizer** (Synth)**:**

```
Original Task:  "[Task Description D]"
After collaborative thinking, the final individual
perspectives from M=[Number of Agents] agents are:
Agent 1:  "[x_1^{(T-1)}]"
...
Agent M: "[x_M^{(T-1)}]"
The final collective understanding synthesized during their
collaboration is:
"[μ^{(T-1)}]"
Based on all this information, please generate a
comprehensive, high-quality, and coherent final solution to
the original task.
```

This prompt structure ensures the Synthesizer has all necessary context to perform its role effectively.

## C  Experimental Setup: In-Depth Information

### C.1  Detailed Benchmark Descriptions

**LiveBench [White et al., 2025]**    LiveBench serves as a dynamic and robust benchmark for evaluating LLM capabilities, characterized by its frequent updates (monthly) to minimize test data contamination and its focus on objectively scorable, challenging tasks. It draws from established hard benchmarks like Big-Bench Hard and AMPS, as well as introducing novel problems. The tasks span a broad range of domains, including:

- *Mathematics:* Encompassing competitive programming problems, olympiad-level mathematics, and algebraic simplification.
- *Reasoning:* Covering logical deduction and spatial reasoning.
- *Language:* Focusing on nuanced understanding and manipulation.
- *Instruction Following:* Testing adherence to complex instructions
- *Data Analysis:* Requiring structured data manipulation

**CommonGen-Hard [Madaan et al., 2023]** CommonGen-Hard, an extension of the CommonGen dataset [Lin et al., 2020], is specifically designed to impose high cognitive load by increasing element interactivity. The core task is to generate a coherent, multi-sentence paragraph incorporating a small set of 3-5 target concepts. The difficulty is amplified by including a large number (approximately 30) of irrelevant distractor concepts from which the model must select and use only the targets, while maintaining narrative coherence and commonsense plausibility. Given its generative nature, evaluation employs an LLM-based evaluator (`gemini-1.5-pro`) guided by a detailed rubric assessing ten dimensions. These dimensions are: (1) **Relevance to Query** (appropriateness and focus, highest weight); (2) **Conciseness** (brevity without losing essential content); (3) **Clarity & Understandability** (ease of comprehension); (4) **Readability & Fluency** (natural language flow, grammatical correctness); (5) **Comprehensiveness & Completeness** (addressing all prompt aspects); (6) **Demonstrated Knowledge** (accurate commonsense or domain knowledge); (7) **Logic & Coherence** (internal consistency and logical structure); (8) **Originality & Creativity** (novelty in ideas or framing); (9) **Engagement & Interest** (compelling nature of the response); (10) **Insightfulness & Depth** (analytical richness beyond surface content, lowest weight). Each dimension is scored (e.g., 1-10), and an aggregated total score is used. This setup directly tests the model's ability to manage high element interactivity and filter relevant information, key aspects related to cognitive load.

## C.2 Detailed Baseline Method Descriptions

The baseline methods used for comparison are implemented as follows:

- **Single Agent (Standard Prompt - IO):** The base LLM is given the task instruction directly, without any specialized prompting techniques, serving as a fundamental measure of its raw capability.
- **Single Agent (CoT):** Chain-of-Thought prompting [Wei et al., 2022] is employed, where the LLM is prompted to "think step by step" or provided with few-shot examples demonstrating a reasoning process before arriving at the final answer.
- **Single Agent (Self-Refine - SR) [Madaan et al., 2023]:** This method involves an iterative process ($T = 3$ iterations). The LLM first generates an initial solution. Subsequently, it is prompted to critique its previous output and then to generate an improved version based on that critique.
- **Multi-Agent Debate (MAD) [Liang et al., 2023, Du et al., 2023]:** Multiple LLM agents ($M = 6$) initially generate individual solutions. In subsequent iterative rounds ($T = 3$ total generations), each agent receives the solutions from all other agents from the previous round and is prompted to consider these peer solutions, critique them if necessary, and refine its own solution. The final answer is typically derived from the best-performing agent's output after the debate rounds.
- **Diverse Multi-Agent Debate (DMAD) [Liu et al., 2025b]:** DMAD extends MAD by promoting diverse reasoning methods from the outset. Each agent is assigned a distinct prompting strategy (e.g., standard IO, Chain-of-Thought, Step-Back Prompting) to generate its initial solution, aiming to break "fixed mental sets." These diverse initial solutions are then shared and refined through iterative debate rounds, similar to MAD.

## C.3 General Implementation Details

Experiments were conducted using Python and Google's Generative AI SDK. **LLM API Parameters:** For all baseline methods (IO, CoT, SR) and the initial generation round ($t = 0$) of multi-agent methods (MAD, DMAD, *CoThinker*), the API temperature was set to "0.25" to encourage some diversity. For subsequent iterative rounds ($t > 0$) in *CoThinker*, MAD, and DMAD, the temperature was set to "0.0" and "frequency_penalty" to "0.5" to promote focused refinement and reduce repetition. Other API parameters (e.g., "top_p", "top_k") were left at their default values. Maximum output tokens were set appropriately for each task.

*CoThinker* **Default Configuration:** Unless specified otherwise in ablation studies, *CoThinker* used $M = 6$ agents, $T_{max} = 3$ interaction rounds (initial generation + 2 refinement rounds), a reference set size $N = 3$ (each agent receives messages from 3 peers), and an exploration rate $\beta = 0.3$.

# D Detailed Experimental Results and Ablation Studies

This appendix provides supplementary experimental results, including comprehensive raw scores for all subtasks across various model families and prompting methodologies. Furthermore, it details ablation studies conducted to investigate the sensitivity of model performance to key hyperparameters.

## D.1 Raw Subtask Performance Scores

The subsequent tables (Table 2 through Table 4) itemize the raw performance scores achieved on each subtask. Scores are reported to two decimal places. A hyphen (-) signifies missing or non-numeric data. Each table is dedicated to a distinct base model family.

Table 2: Raw scores for each subtask for `gemini-1.5-flash-8b` models across different prompting methods.

| Subtask | IO | CoT | SR | MAD | DMAD | CoThinker |
|---|---|---|---|---|---|---|
| Connections | 13.50 | 18.17 | 17.33 | 17.67 | 17.00 | 19.33 |
| CTA | 54.00 | 50.00 | 30.00 | 48.00 | 52.00 | 54.00 |
| Math Comp. | 26.09 | 23.91 | 21.74 | 28.26 | 30.43 | 26.09 |
| Olympiad | 23.82 | 27.64 | 23.84 | 28.25 | 25.87 | 29.00 |
| Paraphrase | 74.27 | 72.82 | 38.42 | 65.22 | 66.55 | 46.02 |
| Simplify | 70.33 | 70.70 | 62.78 | 63.88 | 61.08 | 70.25 |
| Spatial | 34.00 | 28.00 | 18.00 | 34.00 | 22.00 | 28.00 |
| Story Gen. | 73.08 | 68.75 | 62.92 | 66.75 | 67.00 | 65.08 |
| Summarize | 69.35 | 71.27 | 50.43 | 58.32 | 62.62 | 42.32 |
| Table Join | 5.44 | 4.10 | 0.00 | 2.00 | 1.78 | 12.02 |
| Table Reformat | 80.00 | 82.00 | 36.00 | 38.00 | 50.00 | 60.00 |
| Zebra Puzzle | 16.00 | 22.25 | 17.25 | 22.75 | 17.00 | 25.75 |

Table 3: Raw scores for each subtask for `gemini-1.5-flash` models across different prompting methods.

| Subtask | IO | CoT | SR | MAD | DMAD | CoThinker |
|---|---|---|---|---|---|---|
| Connections | 28.17 | 24.00 | 22.83 | 33.17 | 28.50 | 33.67 |
| CTA | 56.00 | 56.00 | 36.00 | 56.00 | 54.00 | 52.00 |
| Math Comp. | 41.30 | 39.13 | 39.13 | 41.30 | 41.30 | 41.30 |
| Olympiad | 32.20 | 34.37 | 33.35 | 34.41 | 33.27 | 36.89 |
| Paraphrase | 80.70 | 78.17 | 52.22 | 80.58 | 82.22 | 72.35 |
| Simplify | 75.83 | 77.68 | 67.57 | 72.07 | 74.40 | 69.00 |
| Spatial | 50.00 | 50.00 | 36.00 | 58.00 | 52.00 | 52.00 |
| Story Gen. | 76.25 | 77.50 | 57.92 | 60.75 | 80.75 | 79.50 |
| Summarize | 77.55 | 75.92 | 54.05 | 68.47 | 74.33 | 68.97 |
| Table Join | 21.64 | 22.78 | 8.12 | 15.00 | 32.60 | 31.20 |
| Table Reformat | 86.00 | 80.00 | 44.00 | 48.00 | 44.00 | 50.00 |
| Zebra Puzzle | 28.50 | 32.00 | 32.50 | 34.25 | 37.50 | 38.50 |

## D.2 Subtask Descriptions

The evaluation benchmark comprises a diverse array of subtasks, each designed to assess specific reasoning and generation capabilities of the models. Concise descriptions for each subtask category are provided below:

**Connections**: Assesses the model's aptitude for identifying and comprehending relationships (e.g., logical, causal, shared attributes) between disparate textual elements or conceptual ideas.
**CTA (Call to Action)**: Evaluates the model's effectiveness in generating or interpreting persuasive or directive language aimed at eliciting a targeted response or action.

Table 4: Raw scores for each subtask for `gemini-1.5-pro` models across different prompting methods.

| Subtask | IO | CoT | SR | MAD | DMAD | CoThinker |
|---|---|---|---|---|---|---|
| Connections | 31.17 | 36.50 | 35.17 | 44.67 | 44.50 | 46.00 |
| CTA | 56.00 | 58.00 | 36.00 | 56.00 | 60.00 | 58.00 |
| Math Comp. | 47.83 | 36.96 | 45.65 | 54.35 | 56.52 | 56.52 |
| Olympiad | 51.79 | 54.77 | 50.16 | 59.63 | 58.46 | 62.72 |
| Paraphrase | 75.37 | 73.78 | 34.18 | 48.50 | 73.88 | 65.17 |
| Simplify | 74.77 | 75.72 | 54.48 | 55.43 | 72.88 | 66.37 |
| Spatial | 44.00 | 48.00 | 36.00 | 34.00 | 38.00 | 38.00 |
| Story Gen. | 69.72 | 68.05 | 42.55 | 56.85 | 67.30 | 73.05 |
| Summarize | 68.92 | 67.17 | 46.23 | 52.83 | 69.05 | 65.72 |
| Table Join | 35.98 | 32.56 | 16.16 | 43.82 | 42.32 | 44.18 |
| Table Reformat | 88.00 | 88.00 | 28.00 | 28.00 | 86.00 | 78.00 |
| Zebra Puzzle | 39.00 | 35.75 | 40.75 | 41.00 | 42.25 | 44.50 |

**Math Comp. (Mathematical Computation)**: Measures the model's proficiency in executing mathematical calculations and resolving problems necessitating computational procedures.

**Olympiad**: Challenges the model with highly complex mathematical problems, characteristic of mathematics Olympiads, which demand profound reasoning and multi-step solution strategies.

**Paraphrase**: Tests the model's ability to accurately rephrase given text while preserving its original semantic content, thereby demonstrating linguistic understanding and versatility.

**Simplify**: Assesses the model's capacity to transform complex textual information into a more readily understandable format, typically by employing simpler vocabulary and sentence structures without loss of core meaning.

**Spatial**: Evaluates the model's spatial reasoning faculties, including its ability to understand and reason about objects in two or three-dimensional space, their interrelations, positions, and transformations.

**Story Generation**: Measures the model's creative ability to produce coherent, engaging, and contextually relevant narratives derived from specified prompts or constraints.

**Summarize**: Assesses the model's proficiency in condensing extended passages of text into succinct summaries that encapsulate the principal points and essential information.

**Table Join**: Evaluates the model's comprehension of relational data structures by requiring it to identify appropriate mechanisms for combining or linking multiple data tables based on common columns or keys.

**Table Reformat**: Tests the model's capability to manipulate tabular data by converting a table from one structural or data representation format to another, adhering to provided instructions.

**Zebra Puzzle**: Assesses the model's deductive reasoning and constraint satisfaction abilities through logic puzzles (such as Einstein's Puzzle) that necessitate deriving a solution from a given set of clues.

### D.3 Ablation Study: Impact of Reference Set Size (N)

This study investigates the influence of varying the reference set size (hyperparameter N) on model performance across selected subtasks. N dictates the number of prior examples or "thoughts" considered by the model during generation. Values of N from 0 (representing a baseline, e.g., standard CoT where N/A) to 5 were evaluated using the `gemini-1.5-flash-8b` model. The results are illustrated in Figure 5.

**Analysis of Figure 5:**

- The general trend in performance on these reasoning-intensive ('olympiad', 'spatial', 'zebra_puzzle') and language-based ('connections') tasks is examined to determine if it improves, plateaus, or reveals an optimal N value.

- Performance at N=0 (baseline) is contrasted with N>0 configurations to ascertain whether the introduction of a reference set confers a tangible advantage for these specific tasks.

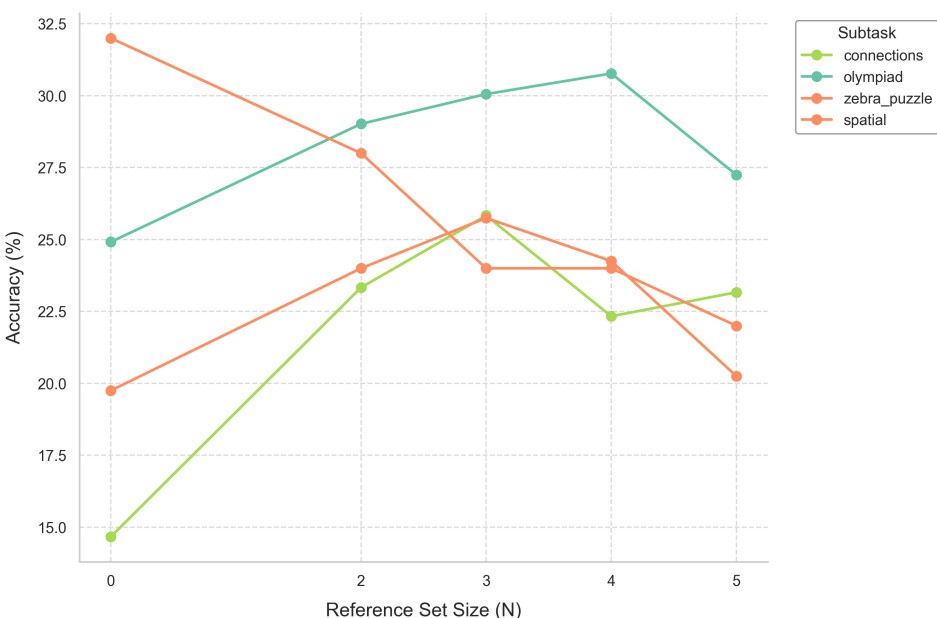

Figure 5: Effect of Reference Set Size (N) on performance for selected subtasks ('connections', 'olympiad', 'zebra_puzzle', 'spatial') using the `gemini-1.5-flash-8b` model. Subtasks are color-coded by their primary category.

- The differential sensitivity of subtasks to variations in N is analyzed, particularly for computationally demanding tasks like 'olympiad' (Math) or 'zebra_puzzle' (Reasoning) relative to 'connections' or 'spatial'.
- The investigation seeks to identify if a particular N value (e.g., N=2 or N=3) consistently yields superior scores or an advantageous performance-cost balance across these subtasks.
- Evidence for diminishing returns is sought, where increasing N beyond a certain point might lead to marginal gains or even performance degradation, potentially due to the introduction of noise or distracting elements from an overly large reference set.

*Contextual Note:* Reasoning and mathematical tasks are often hypothesized to benefit from a moderately sized, diverse reference set. While N=0 or N=1 might provide insufficient context, excessively large N values could introduce irrelevant information.

### D.4 Ablation Study: Impact of Exploration Rate (Beta)

This ablation study explores the effect of the exploration rate (hyperparameter Beta) on model performance for selected subtasks, maintaining a fixed reference set size of N=2. Beta influences the diversity of thoughts or solutions generated by the model. The `gemini-1.5-flash-8b` model was employed for this analysis (Figure 6).

**Analysis of Figure 6:**

- The analysis aims to identify an optimal or effective range for Beta where performance peaks for the selected subtasks, which include data analysis ('tablejoin'), instruction following ('story_generation', 'simplify'), and mathematical computation ('math_comp').
- The impact of extreme Beta values (both very low, indicating minimal exploration, and very high, indicating extensive exploration) on performance is examined for potential suboptimality.
- Differential responses to Beta across subtasks are investigated, for instance, whether creative tasks like 'story_generation' benefit from a different Beta regime compared to more structured tasks such as 'math_comp' or 'tablejoin'.

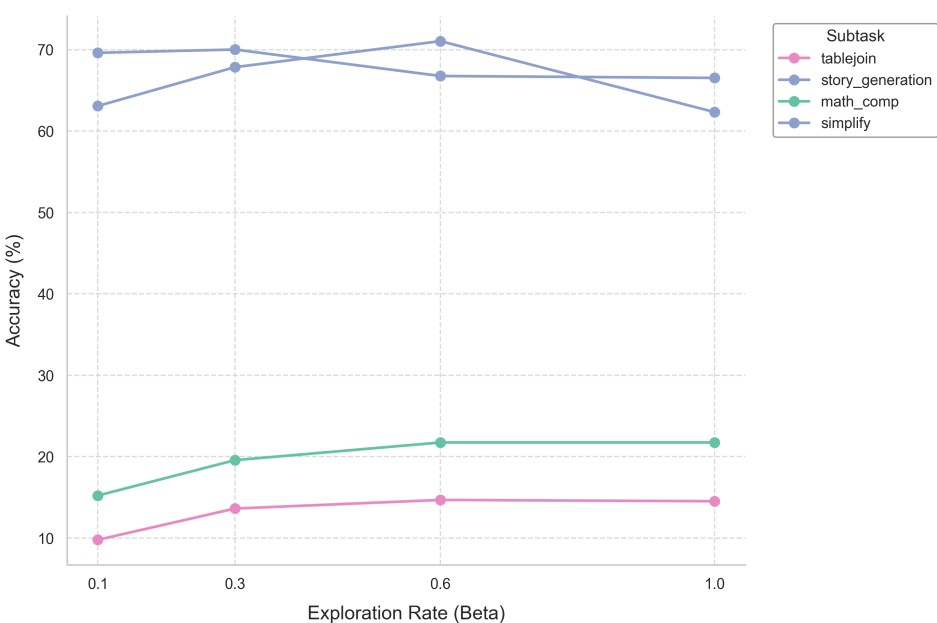

Figure 6: Effect of Exploration Rate (Beta) on performance for selected subtasks ('tablejoin', 'story_generation', 'math_comp', 'simplify') using `gemini-1.5-flash-8b` with N=2. Subtasks are color-coded by their primary category.

- The stability of performance across the spectrum of Beta values is assessed, noting any significant fluctuations versus relatively consistent scores within particular ranges.

*Contextual Note:* A moderate Beta value (e.g., 0.3-0.6 in analogous systems) often represents a balance. Excessively low Beta values might risk premature convergence on suboptimal solutions, while overly high values could lead to an excessively diverse, and potentially lower-quality, set of outputs.

## D.5   Ablation Study: Impact of Number of Agents (M)

This study assesses the influence of the number of agents (hyperparameter M) on performance across all subtasks, with the reference set size fixed at N=3. M denotes the number of independent reasoning paths or "thinkers" utilized by the model. The `gemini-1.5-flash-8b` model was used for this evaluation (Figure 7).

**Analysis of Figure 7:**

- The overall impact of increasing M on performance is analyzed to determine if it generally leads to improvements across most subtasks or if the effects are heterogeneous.
- A cost-benefit perspective is considered, as higher M values, while potentially enhancing performance, also incur increased computational overhead. The study seeks an M value that offers a good trade-off.
- Subtasks that derive particular benefit from a larger number of agents are identified; for example, complex reasoning tasks or those requiring diverse perspectives might exhibit more substantial gains.
- The analysis looks for a saturation point where the benefits of increasing M diminish or where performance might even degrade for some (or all) tasks.

*Contextual Note:* Employing a greater number of agents can enhance the robustness and breadth of exploration. However, an excessive number might not yield significant incremental value or could potentially introduce noise if the aggregation of outputs from multiple agents is not optimally managed.

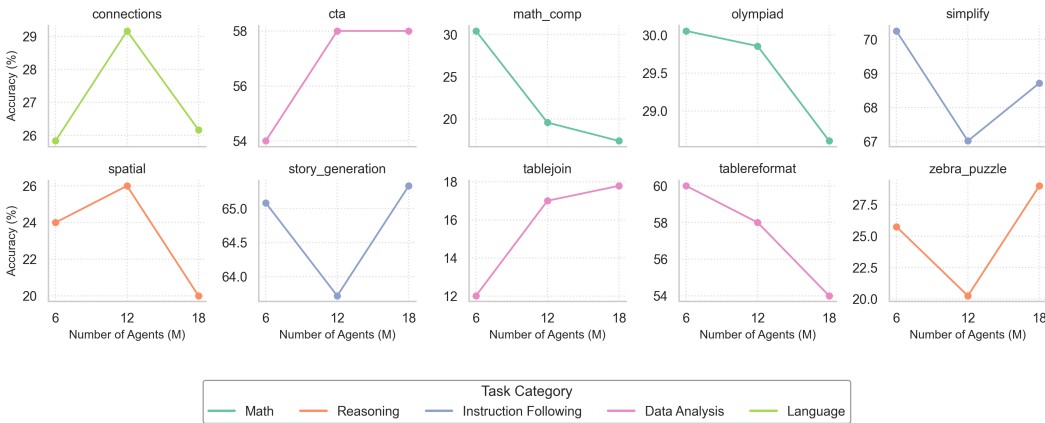

Figure 7: Effect of Number of Agents (M) on performance across all subtasks for `gemini-1.5-flash-8b` with N=3. Each facet corresponds to a subtask, color-coded by its primary category.

### D.6 Ablation Study: Performance for Specific M/N

This analysis evaluates performance across three distinct (M, N) configurations for the `gemini-1.5-flash-8b` model: M6_N3, M12_N6, and M18_N3. These evaluations are conducted under the "With Style" configuration, with Beta fixed at 0.3 and T (temperature or trials) at 3. Results are presented in Figure 8.

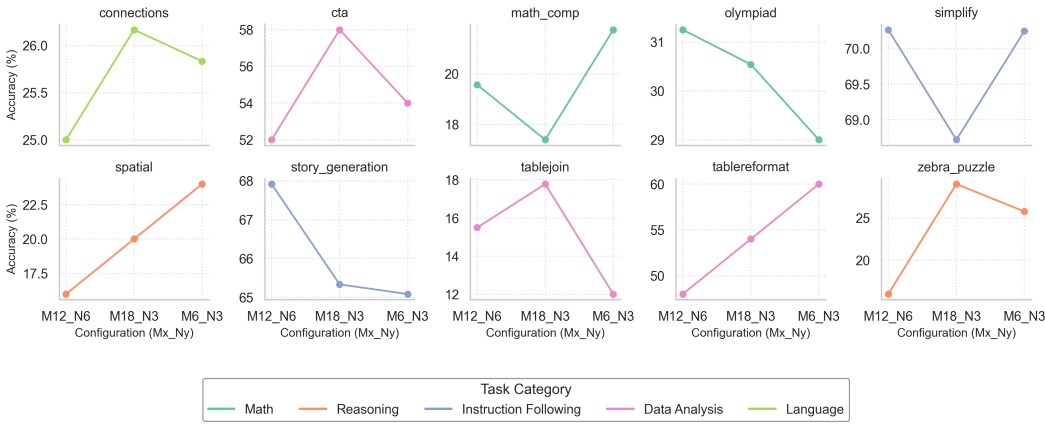

Figure 8: Subtask performance for specific M/N configurations (M6_N3, M12_N6, M18_N3) using `gemini-1.5-flash-8b` under the configuration (Beta=0.3, T=3). Faceted by subtask.

**Analysis of Figure 8:**

- The investigation aims to identify which of the tested (M, N) pairs yields the most favorable performance, either broadly across subtasks or for specific, critical subtasks.

- The trade-off between computational cost and performance gain is considered, as the configurations (M6_N3, M12_N6, M18_N3) entail different computational demands.

- The interaction between M and N is observed by comparing configurations; for instance, whether simultaneous increases in M and N (e.g., M6_N3 to M12_N6) lead to consistent

improvements. The M18_N3 configuration provides insight into a different scaling strategy (higher M, moderate N).

- Consistency in the ranking of these (M, N) configurations across different subtasks is examined.

*Contextual Note:* This study assists in identifying potentially effective fixed configurations by exploring varied scaling strategies for the hyperparameters M and N within the "With Style" framework.

# E    Limitations and Future Work

While *CoThinker* demonstrates promising results in managing cognitive load and enhancing collaborative LLM performance, this work has several limitations that also point towards avenues for future research.

**Limitations** include the scope of LLM evaluation, which primarily utilized models from the Gemini family. The generalizability of specific performance benefits and optimal hyperparameter settings across a wider range of LLM architectures requires further exploration. Additionally, while we argue that *CoThinker* manages transactional costs associated with multi-agent collaboration, a more fine-grained quantitative analysis of these costs versus the gains in solution quality would offer a more complete efficiency profile. The "thinking styles" currently rely on an LLM orchestrator and base styles; the true emergent specialization and their direct impact on distributing intrinsic load warrant deeper investigation.

**Future Work** could explore several promising directions. Developing **adaptive *CoThinker* architectures** that dynamically adjust parameters (number of agents, communication topology) based on real-time task assessment is a key area. **Deeper integration of CLT principles**, such as explicitly modeling and minimizing extraneous load from prompt design or fostering germane load via sophisticated scaffolding, could further enhance performance. Creating methods for **explainability of collective cognition** within *CoThinker*—tracing information flow, identifying critical contributions, and characterizing shared understanding evolution—would improve transparency. Extending the framework for **human-AI collaboration**, incorporating human users as specialized agents, could lead to powerful human-LLM group cognition. Finally, the prospect of such fused intelligence necessitates proactive examination of its **societal implications**, including equity, potential for misuse, accountability, and ethical considerations, demanding robust frameworks for responsible development and governance. Addressing these limitations and pursuing these future directions will further advance our understanding of how to build truly collaborative and cognitively capable LLM-based systems.

