# OpenReview forum: "United Minds or Isolated Agents? Exploring Coordination of LLMs under Cognitive Load Theory"
_NeurIPS.cc/2025/Conference — Submitted to NeurIPS 2025_

### Official Review · Reviewer_bai9 · 2025-06-16

**Clarity:** 3
**Significance:** 2
**Originality:** 3
**Rating:** 4
**Confidence:** 4

**Summary:**

This paper introduces CoThinker, a novel multi-agent Large Language Model (LLM) framework that draws a strong analogy from Cognitive Load Theory (CLT) in human cognitive science to explain and mitigate performance ceilings observed in LLMs on complex, multi-faceted tasks. The authors posit that these limitations arise when task demands exceed an LLM's effective cognitive load capacity, similar to human cognitive overload. CoThinker operationalizes CLT principles by distributing intrinsic cognitive load through agent specialization (Agent Parallel Thinking), managing transactional load via structured communication (Communication Moderator), and employing a collective working memory (Transactive Memory System).

**Questions:**

Just out of curiosity, how does the theoretical "cognitive load ceiling" you propose relate to or interact with the practical limitations imposed by context windows in LLMs? Is there a direct correlation (i.e., does exceeding the context window automatically imply cognitive overload), or are these distinct but possibly related constraints?

**Ethical Concerns:**

["NO or VERY MINOR ethics concerns only"]

**Final Justification:**

Authors have addressed my initial concerns, especially in clarifying the architectural differences from existing multi-agent frameworks and providing additional experiments to demonstrate the effectiveness of this framework on other LLMs, thus validating it to be a model-agnostic approach.

**Limitations:**

yes, in Appendix E

**Paper Formatting Concerns:**

nothing stands out to me

**Quality:**

3

**Strengths And Weaknesses:**

**Strengths**
1. The paper's core strength lies in its innovative approach of drawing inspiration from Cognitive Load Theory (CLT) to understand and address LLM performance limitations. This analogy between human cognitive load and LLM capacity provides a credible and well-grounded theoretical foundation for the CoThinker architecture.
2. The CoThinker design is logically structured, with each module (Agent Parallel Thinking, Transactive Memory System, Communication Moderator, Synthesizer) directly motivated by principles from CLT and human collective intelligence. This theoretical grounding makes the design choices seem principled and effective.
3. The empirical performance is decent. Also, the inclusion of interesting ablation studies further strengthens the paper by providing insights into the impact of key architectural parameters and their relation to cognitive load management.
4. It is interesting to see systematical operationalization of CLT within a multi-agent LLM framework

**Weaknesses**
1. While Section 2 critiques limitations of existing multi-agent designs in their motivation, it lacks a detailed architectural comparison with prominent existing frameworks (e.g., AutoGen). This makes it challenging to clearly discern CoThinker's unique architectural contributions and how its specific design choices directly overcome the architectural flaws, rather than just motivational shortcomings, of prior work. A more explicit comparative analysis of architectural elements would clarify the work's distinct advancements.
2. A critical limitation is the exclusive focus on the Gemini series of LLMs for experimentation. As noted, Table 1 reveals that the comparative performance of different architectures varies across different Gemini models (e.g., gemini-1.5-flash vs. gemini-1.5-flash-8b), suggesting model-specific sensitivities. This raises significant questions about the generalizability of CoThinker's observed benefits and optimal parameter settings to other LLM families (e.g., OpenAI's GPT series, Claude, Llama, Qwen etc). The paper's conclusions might be constrained to the specific characteristics of Gemini models.

Minor suggestions:

i. While the paper provides an analysis of LiveBench results, a more detailed breakdown for Table 1 is needed. For instance, for gemini-1.5-flash and gemini-1.5-pro models, CoThinker is only slightly better than DMAD on average. Conversely, for gemini-1.5-flash-8b, CoThinker shows a more substantial improvement over DMAD but is only slightly superior to CoT. A deeper, more granular discussion on why these varying degrees of improvement are observed across different model capacities and baselines would significantly enrich the analysis and demonstrate a more thorough understanding of the results.

ii. In the main text, I would suggest authors to specify which part of Appendix are referred rather than claiming generally "See details in Appendix"

---

> ### Author Rebuttal · Authors · 2025-07-30
>
> Thank you for your careful reading of our paper and for the insightful feedback. We clarify two points upfront:
>
> 1. Our work is proposed not as a competing platform to frameworks like AutoGen, Agentverse or Chatdev, but as a complementary set of 'plug-and-play' cognitive principles designed to enhance existing systems.
> 2. Our framework is designed to be model-agnostic: any LLM with certain social behaviors should benefit from it.
>
> We have tried to address all of your concerns below and everything will be added into the final version of manuscript.
>
> **Response to Question: Cognitive Load Ceiling vs. Context Window**
>
> *Just out of curiosity, how does the theoretical "cognitive load ceiling" you propose relate to or interact with the practical limitations imposed by context windows in LLMs?*
>
> It's an interesting question touching the heart of our theoretical framework. We'll answer it first to clarify foundational concepts.
>
> 1.  The **cognitive load ceiling** is typically reached *long before* the context window is filled, especially for tasks with high element interactivity (e.g., a reasoning task requiring tracking logical steps throughout a long prompt). This overload isn't due to the "physical" size of the context but to the limitations of the attention mechanism. When a task requires the model to process and integrate many interdependent pieces of information, the required attention becomes too scattered across tokens. This can lead to "drifting attention" and an inability to complete the task successfully (*We've added a section regarding Theoretical & Empirical Justification of LLM Working Memory at the end for your interest*).
> 2. The **context window** is a "physical" limitation of the model as models are not trained beyond it. Thus the model's attention will naturally drift over the context window, even when the task is easy. Our theoretical framework well explains the issue of reaching context window limits, and we also correct the misconception that the context window as the LLM's working memory.
>
> **Response to Weakness 1: Lack of Architectural Comparison**
>
> *...it lacks a detailed architectural comparison ... discern CoThinker's unique architectural contributions...*
>
> We provide a detailed comparison below. **AutoGen** is designed as an engineering platform for building multi-agent systems, aggregating functional components to meet a goal. Its design choices reflect this engineering purpose: agent roles are often functional (e.g., `Coder`, `Planner`), communication is frequently managed via a broadcast-style `GroupChatManager`, and memory is typically a coarse-grained conversation history.
>
> In contrast, **CoThinker** provides a set of **cognitive design principles** derived from CLT, aimed at boosting the efficiency of agent collaboration itself. Our approach isn't a competing platform but rather an exploration of a new, complementary dimension of MAS design. These principles could, be integrated into frameworks like AutoGen to enhance their collaborative core.
>
> |Feature| **CoThinker** (Cognitive Design)| **AutoGen** (Engineering Platform)|
> |:-|:-|:-|
> |Agent Specialization|Dynamic Thinking Styles assigned based on the task to distribute cognitive labor.| Functional Roles (e.g., `CodeAgent`, `UserProxyAgent`) defined for engineering purposes.|
> |Communication|Communication Moderator balance focused refinement and diverse exploration.| Broadcast Group Chat or pre-defined conversational flows for programmatic interaction.|
> |Memory & State|Structured Transactive Memory System (TMS) that synthesizes a "collective mind" (consensus, divergence, expertise).| Coarse-Grained Conversation History, often relying on external databases for long-term memory.|
>
> In short, our work serves a different purpose. We are not competing with platforms like AutoGen but are instead proposing a cognitively-grounded approach to enhance agent cooperation, moving beyond purely functional roles.
>
> **Action:** We'll add this detailed architectural comparison to **Section 2** of the revised manuscript.
>
> ---
>
> **Clarification to Weakness 2: Exclusive Focus on Gemini Models**
>
> *... focus on the Gemini series... generalizability...*
>
> Our core thesis is that **CoThinker is a model-agnostic framework that coordinates a model's inherent abilities to reduce cognitive load**. Our methods leverage emergent social behaviors possessed by the LLM rather than relying on its internal parameters. Therefore, any LLM exhibiting these general interactive capabilities should benefit.
>
> For your information, we conducted new experiments across other model families. The results validate our thesis, showing consistent improvements. The enhancement is particularly significant for models with stronger reasoning capabilities, highlighting our work as a powerful amplifier for existing intelligence.
>
>
> |Model|Method|Avg. Score (LiveBench)|Math Score|Reasoning Score|
> |:-|:-|:-:|:-:|:-:|
> |Gemini 2.5 Flash|CoThinker| **72.8**|**76.3**|**69.2**|
> ||DMAD|59.7|56.7|62.8|
> ||IO (Baseline)|45.1|59.3|31.0|
> |GPT-4.1-mini|CoThinker|**55.4**|**40.0**|**70.8**|
> ||DMAD|39.1|34.2|44.0|
> ||IO (Baseline)|37.4|34.0|40.8|
> |Qwen3-32B|CoThinker|**22.1**|**18.9**|**25.2**|
> ||DMAD|11.5|8.8|14.2|
> ||IO (Baseline)|11.7|3.4|20.0|
> |DeepSeek-R1 (qwen-8B)|CoThinker|**5.8**|**2.9**|**8.8**|
> ||DMAD|5.2|3.8|6.5|
> ||IO (Baseline)|2.3|1.9|2.8|
>
>
>
> **Response to Minor Suggestions**
>
> *i. ...a more detailed breakdown for Table 1 is needed... A deeper, more granular discussion...*
>
> The aggregated performance in Table 1 can be misleading, as it includes tasks with low intrinsic cognitive load (e.g., Instruction Following). On these tasks, the overhead of a multi-agent framework can slightly lower the relative score. This is fully consistent with CLT. The true strength of CoThinker is revealed in high-load domains.
>
> **Action:** To provide this granular analysis, we will revise **Section 5.2** to discuss this nuance. Furthermore, we'll add **qualitative case studies** to the appendix, presenting a fine-grained comparison of agent outputs on a specific reasoning task.
>
> ---
> **Theoretical Justification of LLM working memory**
> Here we provide details responding to the concern about *the lack of formal proof or empirical evidence linking CLT to specific internal representations or mechanisms within LLMs*.
> 1. We first identify the key *characteristics* of working memory. We then identify the *dual characteristics* in LLMs that corresponding to the working memory features.
> 2. Finally, we provide empirical evidence to support the existence of these dual characteristics in LLMs.
>
> **1. Key Characteristics of Working Memory**
> By definition, working memory is a feature that temporarily holds and processes information simultaneously. By definition, cognitive load is the *attention* needed for interactive information within working memory (mental effort), which determines the *easiness* of task completion [1].
>
> Thus, the key characteristics to quantify working memory are the "attention on interactivity of information" and the "easiness of task completion". Interestingly, the *attention mechanism* in LLMs and its output *perplexity* naturally becomes the dual characteristics providing a way to quantify these two aspects.
>
> **2. Pre-experiment 1: Attention as a proxy for cognitive load**
> 1. **Justification**:
> $$
> \text{Attention Entropy} = -\sum_{i=1}^{N} a(s_i) \log a(s_i)
> $$
> measures diversity of attention distribution across different tokens. A higher entropy indicates a more uniform distribution of attention, suggesting the model is considering multiple aspects of the input, which corresponds to a higher cognitive load in WM [2].
>
> 1. **Experiment and results**: We choose a set of Q&A pairs and construct a 4 level difficulty question set. We constrol the length of the Q&A for fair comparison. We then measure the average entropy of the attention distribution of each layer in the model. The results are shown in the table below:
>
> | Task Complexity Level | Avg. Attention Entropy (No Reasoning) | Avg. Attention Entropy (With Reasoning) |
> |:-|:-:|:-:|
> |Level 1|4.442|4.439|
> |Level 2|4.796|4.726|
> |Level 3|5.043|4.937|
> |Level 4|6.101|5.920|
>
> 1. **Implication**: The attention entropy increases with task complexity, indicating that the model have to use more pieces of information to process the task, which corresponds to higher cognitive load. The presence of reasoning steps consistently lowers attention entropy, indicating that reasoning helps the model to "offload" some of the cognitive load by focusing its attention on key pieces of information.
>
> **3. Pre-experiment 2: Perplexity as a proxy for working memory**
> 1. **Justification**:
> $$
> \text{Perplexity}=\exp\left(-\frac{1}{N}\sum_{i=1}^{N}\log p(s_i)\right)
> $$
> measures the certainty of the model's generation, which means the "easiness" of task completion.
> 1. **Experiment and results**: Full details on its validity are in the Appendix. We choose a set of Q&A pairs (high quality answer) and construct 5 guiding instructions with increasing complexity. We then measure the perplexity of the on the answer for each instruction:
>
> |Instruction Complexity Level|Avg. Perplexity (hard)|Avg. Perplexity (easy)|
> |:--|:-:| :-: |
> |Level 1|120.5013|3.3661|
> |Level 2|88.9650|3.4227|
> |Level 3|85.3538|3.4522|
> |Level 4|92.4836|3.4572|
> |Level 5|100.7143|3.4603|
>
> 1. **Implication**: For hard tasks, perplexity first decreases then increases, indicating that the instructions help the model to focus on the task up to a certain point, and then become an additional burden on the model's cognitive load; For easy tasks, perplexity increases with instruction complexity showing no help. This measurement of "easiness" aligns with CLT.
>
> **Reference**
>
> [1] Sweller, John. "Cognitive load theory." Psychology of learning & motivation, 2011.
>
> [2] Zhang, et al. Attention Entropy is a Key Factor: An Analysis of Parallel Context Encoding with Full-attention-based Pre-trained Language Models. ACL, 2025.

---

> > ### Author Response · Authors · 2025-08-02
> >
> > Dear Reviewer bai9,
> >
> > Thank you again for your thoughtful and constructive review. We have posted a rebuttal aiming to address your questions with a more detailed architectural comparison and new results on generalizability.
> >
> > We would appreciate your feedback to ensure we've resolved these points and are ready to discuss any remaining concerns.
> > Thank you for your valuable time and insights.
> >
> > Best regards,

---

> > ### Comment · Reviewer_bai9 · 2025-08-03
> >
> > Thank you for your detailed responses and additional experiments. My concerns have been addressed. I would raise my rating to 4.

---

> > > ### Author Response · Authors · 2025-08-03
> > >
> > > Dear Reviewer bai9,
> > >
> > > Thank you for your positive feedback. We're glad that our response addressed your concerns.
> > >
> > > All the best,

---

### Official Review · Reviewer_evwp · 2025-06-18

**Clarity:** 2
**Significance:** 3
**Originality:** 3
**Rating:** 4
**Confidence:** 3

**Summary:**

This paper proposes that the performance limitations of LLMs on complex tasks arise from cognitive overload, drawing an analogy with Cognitive Load Theory (CLT) in human cognition. To address this, the authors introduce CoThinker, a multi-agent framework that distributes cognitive load among specialized LLM agents through shared memory and structured communication. Empirical results demonstrate that CoThinker significantly improves performance and reasoning efficiency on high-load tasks compared to existing multi-agent and single-agent baselines.

**Questions:**

- Is the working memory described in Section 3.1 also incorporates LLM’s own parametric knowledge? From the discussion seems the analogy is only related to the in-context information. What’s the position of parametric knowledge in this analogy? Seems the “elicitation” (not memorization) of parametric knowledge won’t be affected by something like LLM’s own capacity.
- How is your theory and agent design related to the concept of “cognitive offloading”? Instead of think in a “parallel” way, it’s more like LLM with self-awareness of its capabilities could offload some burden to other external model ore resources, in oder to cooperate and achieve success.
- For the network construction in method, is it constructed by agent proactively during problem solving, or it is settled before any interaction? Could you provide an example on how the information is generated and passed in the system? For example, does the agent only generates by gathering information from certain agent, or it also proactively broadcast information to certain agent, using TMS as the media?
- Is it possible to mix agents of different possibilities in the same network? Will this make an abolition study and provide insights on each agent’s role?

**Ethical Concerns:**

["NO or VERY MINOR ethics concerns only"]

**Final Justification:**

I think the author's response for most of the concerns are reasonable, and we have thoroughly discussed through the rebuttal about potential weakness and how they could be further addressed. I personally believe additional result from DeepSeek-R1 Distilled Qwen 32B would make the paper's arguments more persuasive if the performance could reach 45% or more. Current arguments are acceptable given the limited time and budget. I decide to maintain my original positive score.

**Limitations:**

Limitations could be discussed from the role of each component in the framework, what are some misaligned aspect between the original CL theory and the LLM side, and what are some error cases that still needs improvements.

**Quality:**

3

**Strengths And Weaknesses:**

Strength:
- The grounding of LLM capability into the cognitive load theory itself is novel and insightful, providing a good motivation for the design in the paper
- The paper’ analysis in agent network is insightful and inspiring, especially the ablation part (Section 5.4)

Weakness:
- The paper’s writing on the network construction part could be further improved and be more concrete. Currently from the writing it’s still not very clear how the information is passed over different agents and how they exactly collaborate in problem-solving
- The model used for testing only involves gemini model which is limited. Also, it’s possible to further conduct ablation study on different LLM configuration and each component’s role. For instance, how would the ability of the synthesizer affect the final agent system’s performance?

---

> ### Author Rebuttal · Authors · 2025-07-30
>
> We appreciate your insightful feedback and are glad you found our application of Cognitive Load Theory (CLT) novel and our analysis inspiring. We address your questions and suggestions below.
> ### Response to Questions
>
> **1. Is the working memory described in Section 3.1 also incorporates LLM’s own parametric knowledge?**
>
> To better answer your question, we first clarify two concepts. An LLM's **parameters** incorporate two aspects: 1) **Parametric Knowledge**, corresponding to human long-term memory, and 2) model's **innate generation capabilities**. In our framework, **Working Memory (WM)** is the mechanism of selective attention—a limited capacity to hold and manipulate information simultaneously—and isn't simply the in-context information itself.
>
> Therefore, a model's parameters determine how effectively it selects information for its WM, while its parametric knowledge affects how well it can understand and integrate that information into its reasoning process (reflected in its attention patterns). WM is the resource used to process information, requiring the LLM to load both context and activated parametric knowledge for a task.
>
> The reason that simple `elicitation` appears unrelated to WM is that it often involves sequentially giving facts, which consumes little WM as it doesn't require integrating many interdependent pieces of information. However, if asked to elicit a large volume of structured information, the LLM would need to use its WM to remember what it has outputted and what comes next, which can indeed lead to cognitive overload.
>
> *(Note: For mathematical justification of LLM WM and its link to attention, please see a note at the end.)*
>
> **2. How is your theory and agent design related to the concept of “cognitive offloading”?**
>
> Our framework aligns with cognitive offloading, through our **Transactive Memory System (TMS)**. The TMS maintains a consensus and "who knows what." We use agents' self-awareness so it can check the TMS to see what cognitive functions other agents have performed (e.g., "Agent B has done the fact-checking"), allowing it to offload that function and focus on other parts of the task.
>
> **3. For the network construction in method... how the information is generated and passed in the system?**
>
> The network updates dynamically each round. To illustrate the information flow, let's take **Agent 1** as an example:
> 1.  The **Communication Moderator** first selects current peers for Agent 1 to interact with, say Agent 2 & 3.
> 2.  Using the **TMS** as a medium, Agent 1 accesses two key pieces of information: the group's current **consensus** and the **"who knows what"**.
> 3.  From the TMS, Agent 1 may learn that Agent 2 is performing fact-checking and Agent 3 is identifying relevant mathematical formulas. Realizing these parts are being handled, Agent 1 **offloads** these tasks. It can read Agent 3's output, take the proposed formulas to find alternative solutions.
> 4.  The new outputs from all agents are sent back to the TMS and the Communication Moderator, informing the network for the subsequent round.
>
> **4. Is it possible to mix agents of different possibilities in the same network?**
>
> Yes, our framework implements a form of this. The **"Agent Parallel Thinking"** module assigns each agent a specific cognitive focus (a "thinking style"), creating a network of agents with different specialized abilities. To demonstrate the effectiveness of this component, we conducted an ablation study (`w/` vs. `w/o` the thinking style module). Below, assigning thinking styles consistently improves performance.
> |Model `gemini-1.5-pro`|Average|Data Analysis|Instruction Following|Language|Reasoning|
> |:-|:-:|:-:|:-:|:-:|:-:|
> |**w/ Styles**|**53.6**|58.0|**65.7**|**46.0**|**44.5**|
> |w/o Styles|51.1|58.0|64.1|41.5|41.0|
>
> *(Note: In `w/o Styles` condition, identical agents were used, which required a non-zero temperature to encourage diversity. This setting gives the baseline an advantage over a purely greedy approach, yet the module still demonstrates clear benefits.)*
>
> **Action:** We'll add this ablation study to appendix, together with **case studies** showing how different agents collaborate.
>
> ---
>
> ### Response to Weaknesses
>
> **1. The paper’s writing on the network construction part could be further improved...**
>
> We'll incorporate detailed information flow provided in our answer to **Question 3** into the main manuscript.
>
> **2. The model used for testing only involves gemini model which is limited. Also, it’s possible to further conduct ablation study on different LLM configuration and each component’s role.**
>
> To clarify, our framework doesn't compete with a model's native ability but to **coordinate it effectively**. As long as model possesses certain emergent social behaviors, our framework can enhance its performance by reducing cognitive load generally. Our approach is designed to be **model-agnostic**, coordinating a model's capabilities. For your information, we have also added results from other model families.
>
>
> |Model|Method|Avg. Score (LiveBench)|Math Score|Reasoning Score|
> |:-|:-|:-:|:-:|:-:|
> |Gemini 2.5 Flash|CoThinker| **72.8**|**76.3**|**69.2**|
> ||DMAD|59.7|56.7|62.8|
> ||IO (Baseline)|45.1|59.3|31.0|
> |GPT-4.1-mini|CoThinker|**55.4**|**40.0**|**70.8**|
> ||DMAD|39.1|34.2|44.0|
> ||IO (Baseline)|37.4|34.0|40.8|
> |Qwen3-32B|CoThinker|**22.1**|**18.9**|**25.2**|
> ||DMAD|11.5|8.8|14.2|
> ||IO (Baseline)|11.7|3.4|20.0|
> |DeepSeek-R1 (qwen-8B)|CoThinker|**5.8**|**2.9**|**8.8**|
> ||DMAD|5.2|3.8|6.5|
> ||IO (Baseline)|2.3|1.9|2.8|
>
> Regarding component ablations, the role of the **Communication Moderator** is analyzed in the paper's main ablation studies (Fig. 4), which investigate its key parameters (`N` and `β`). The ablation on different agents is provided above rebutal and will be added to appendix.
>
> ---
>
> ### Response to Limitations
>
> *Limitations could be discussed from the role of each component...what are some misaligned aspect between the original CL theory and the LLM side, and what are some error cases...*
>
> These are excellent suggestions for a more robust discussion.
>
> **Action:** We'll revise our **Limitations and Future Work** section to specifically include:
> 1.  An analysis of assumptions related to each component's role.
> 2.  A discussion of misalignments in the CLT-LLM analogy.
> 3.  An examination of common error cases to better define boundaries.
>
> ---
>
> **Theoretical Justification of LLM WM**
> Here we provide details responding to your second concern on *the lack of formal proof or empirical evidence linking CLT to specific internal representations or mechanisms within LLMs*.
> 1. We'll first identify the key *characteristics* of WM. We'll then identify the *dual characteristics* in LLMs that corresponding to the WM features.
> 2. Finally, we'll provide empirical evidence to support the existence of these dual characteristics in LLMs.
>
> **1. Key Characteristics of WM**
> By definition, WM is a feature that temporarily holds and processes information simultaneously. By definition, cognitive load is the *attention* required to handle interactive information within WM (mental effort), which determines the *easiness* of task completion [1].
>
> Thus, the key characteristics to quantify WM are the "attention on interactivity of information" and the "easiness of task completion". Interestingly, the *attention mechanism* in LLMs and its output *perplexity* naturally becomes the dual characteristics providing a way to quantify these two aspects.
>
> **2. Pre-experiment 1: Attention as a proxy for cognitive load**
> 1. **Justification**:
> $$
> 	\text{Attention Entropy} = -\sum_{i=1}^{N} a(s_i) \log a(s_i)
> $$
> represents the diversity of attention distribution across different tokens. A higher entropy indicates a more uniform distribution of attention, suggesting model is considering multiple aspects of the input, corresponding to a higher cognitive load in WM [2].
>
> 1. **Experiment and results**: We choose a set of Q&A pairs and construct a 4 level difficulty question set. We constrol the length of the Q&A for fair comparison. We measure average entropy of the attention distribution of each layer in model.
>
> |Task Complexity Level|Avg. Attention Entropy (No Reasoning)|Avg. Attention Entropy (With Reasoning)|
> |:-|:-:|:-:|
> |Level 1|4.442|4.439|
> |Level 2|4.796|4.726|
> |Level 3|5.043|4.937|
> |Level 4|6.101|5.920|
>
> 1. **Implication**: The attention entropy increases with task complexity, indicating that model have to use more pieces of information to process the task, corresponding to higher cognitive load. Adding reasoning steps lowers entropy, indicating that reasoning helps "offload" cognitive load by focusing attention.
>
> **3. Pre-experiment 2: Perplexity as a proxy for WM**
> 1. **Justification**:
> $$
> 	\text{Perplexity}=\exp(-\frac{1}{N}\sum_{i=1}^{N}\log p(s_i))
> $$
> represents model's certainty, measuring the "easiness" of task completion.
> 1. **Experiment and results**: We'll briefly describe the experiment here and details of validity of experiment to be found in Appendix. We choose a set of Q&A pairs and construct 5 guiding instructions with increasing complexity. We then measure the perplexity of the on the answer for each instruction:
>
> |Instruction Complexity Level|Avg. Perplexity (hard)|Avg. Perplexity (easy)|
> |:--|:-:| :-: |
> |Level 1|120.5013|3.3661|
> |Level 2|88.9650|3.4227|
> |Level 3|85.3538|3.4522|
> |Level 4|92.4836|3.4572|
> |Level 5|100.7143|3.4603|
>
> 1. **Implication**: For hard tasks, perplexity initially decreases and then increases, showing that instructions help up to a point before becoming a cognitive burden; While for easy tasks, the perplexity increases with complexity showing no help. This measurement of "easiness" perfectly aligns with the CLT.
>
> **Reference**
>
> [1] Sweller, John. "Cognitive load theory." Psychology of learning & motivation, 2011.
>
> [2] Zhang, et al. Attention Entropy is a Key Factor: An Analysis of Parallel Context Encoding with Full-attention-based Pre-trained Language Models. ACL, 2025.

---

> > ### Comment · Reviewer_evwp · 2025-07-31
> >
> > Thank the author for clarifications on these questions. Here are some further concerns:
> >
> > 1. From your response to Question 2 and 3 it seems that the model's cognitive offload is still triggered by its awareness of what other agents have done. This seems like a passive behavior (more like awareness, corresponding adaptation and decision-making) in cognitive offloading, instead of an active one where the agent proactively offload a task to other specific agents. Correct me if I am wrong. Therefore I am not sure whether the design itself is exactly aligned with the concept of cognitive offloading. Instead I feel pre-experiment you give at the very end seems more like a justification for the analogy you made, but how the multi-agent design adapts it is still a little bit questionable from my view.
> >
> > 2. Further experiments on more models show that current open-source models still significantly under-performs when adapted into your framework. I have checked the current LiveBench leaderboard and seems model like DeepSeek R1 Distill Qwen 32B could achieve over 40% general performance, but seems the 8B model achieves only 5.8%. I am not sure whether the model size (given data trained on is similar) could create such huge gap or if the proposed framework is suitable for current open-sourced models. This further raises some concerns.
> >
> > I think all my other concerns are well-addressed for now and the author gives a very comprehensive and detailed rebuttal with concrete experiment results.

---

> > > ### Author Response · Authors · 2025-08-01
> > >
> > > **Response to Follow-up Question 1 (on Cognitive Offloading):**
> > >
> > > Thank you for this insightful observation. You are right that our framework implements a more reactive form of cognitive offloading, rather than a proactive one via explicit "tool calling." This is a deliberate design choice grounded in established theory of Cognitive Offloading.
> > >
> > > As detailed in the literature on **Cognitive Offloading** [3], one of the key mechanisms for offloading in social contexts is the **Transactive Memory System (TMS)**. Our framework is designed to model precisely this. Within a TMS, offloading is often a reactive process: an agent becomes aware of the group's "Specialization of Knowledge" (i.e., seeing what others have done) and adapts its own cognitive focus accordingly, thereby reducing its own load.
> > >
> > > Therefore, our system's behavior is a direct implementation of this core aspect of social cognitive offloading. We agree that exploring more proactive delegation mechanisms is an excellent and inspiring direction for future work.
> > >
> > > ---
> > > **Response to Follow-up Question 2 (on Open-Source Model Performance):**
> > >
> > > We appreciate you raising this point about absolute performance. We offer two key clarifications:
> > >
> > > 1.  **Strong Relative Improvement:** The primary goal of our framework is to enhance a model's inherent capabilities by reducing cognitive load. Our results show it is highly effective in this regard. For instance, with DeepSeek-R1 8B, CoThinker **more than doubles** the baseline performance (from 2.3% to 5.8%). This strong relative gain demonstrates our framework's value, even when the base model has limitations.
> > >
> > > 2.  **Impact of Decoding Strategy:** The performance gap with the LiveBench leaderboard is explained by a crucial methodological difference. For reproducibility, our experiments use a **greedy decoding** strategy. In contrast, leaderboards often use sampling-based methods, which are known to yield higher scores but are less deterministic. This effect is particularly pronounced for certain open-source models. Our results therefore reflect the framework's robust performance under a stricter evaluation setting.
> > >
> > > [3] Risko, Evan F., and Sam J. Gilbert. "Cognitive offloading." Trends in cognitive sciences 20, no. 9 (2016): 676-688.

---

> > > > ### Comment · Reviewer_evwp · 2025-08-01
> > > >
> > > > I think the author's response for these concerns are reasonable, even though for the second concern adding an additional result from DeepSeek-R1 Distilled Qwen 32B would make your arguments more persuasive if the performance could reach 45% or more. Current arguments are acceptable given the limited time and budget. Currently I have no further questions. Hope the additional results, discussion, and arguments could be incorporated in the next revision to further improve the quality of this paper. Thanks the author for the response.

---

> > > > > ### Author Response · Authors · 2025-08-02
> > > > >
> > > > > Dear Reviewer evwp,
> > > > >
> > > > > Thank you for your response and for the insightful follow-up discussion. We appreciate the suggestion and will consider adding new test on DeepSeek-R1 Distilled Qwen 32B in our final . We will be sure to incorporate above additional results and this valuable discussion into the final version to further strengthen the paper.
> > > > >
> > > > > Wishing you all the best,

---

> > > > > ### Author Response · Authors · 2025-08-05
> > > > >
> > > > > Dear Reviewer evwp,
> > > > >
> > > > > We are writing to gently follow up and see if you have had a chance to submit your final rating.
> > > > >
> > > > > Thank you once again for your detailed feedback and for the constructive discussion. We were very encouraged that you found our responses to your concerns reasonable. We hope that the paper is now considered a stronger contribution.
> > > > >
> > > > > We will be sure to incorporate all of your valuable suggestions into the final revision.
> > > > >
> > > > > Best regards,
> > > > > The Authors

---

### Official Review · Reviewer_KZNb · 2025-06-30

**Clarity:** 4
**Significance:** 3
**Originality:** 3
**Rating:** 5
**Confidence:** 4

**Summary:**

The authors address a failure mode of LLMs when presented with complex tasks. Specifically, tasks that require integration of large amounts of information. They draw inspiration from Cognitive Load Theory in cognitive science in order to analyze this scenario and hipothesize that LLMs' failures are mainly due to an excess of cognitive load. Following the same analytical framework, the authors propose an architecture aimed at lowering the cognitive load of LLMs upon collaboration, and they test its impact on two complex benchmarks. The authors also compare the performance of LLMs under the proposed architecture vs other 5 single- and multi-agent baselines.

**Questions:**

I have the following questions / suggestions:
1. What were the motivations or selection criteria for the evaluation benchmarks? I can guess roughly from the rest of the paper, but it would be nice to have it explicitly estated, maybe also referencing one or two benchmarks that might seem suitable to inexpert eyes (this last bit is just a suggestion, so highly optional).
2. You only evaluate gemini models. Could you elaborate on that decision and how these results might or might not transfer to other model families?
3. The limitations addressed in the appendix should be in the main paper. I also believe that the future work is always inspiring and should be addressed in the main paper.

**Ethical Concerns:**

["NO or VERY MINOR ethics concerns only"]

**Final Justification:**

I stand by my previous rating, but I do acknowledge that the paper has improved after the authors' additions and clarifications. Most importantly, authors have elaborated on their motivations for choosing the selected benchmarks, they have included new evaluations for different models, and have included the limitations in the main paper, as suggested in my review. Other concerns have been properly discussed and justified.

**Limitations:**

Addressed, but only in the appendix. Should be included in the main paper.

**Quality:**

3

**Strengths And Weaknesses:**

Strenghts:
- The inspiration from CLT in cognitive science is solid, highly interesting and has potential.
- Revision of related work seems thorough.
- Claims are clear.
- Explanation of the cognitive background and proposed method is very clear and easy to follow, including figures and formal descriptions when needed.
- Empirical evaluation seems correct, and the use of relative metrics makes the comparison against the baselines easier.
- Good analysis of results and ablation studies.

Weaknesses:
- Reasons for the selection of the two benchmarks are a bit scarce. Could elaborate more on the motivations and criteria for their selection.
- Only evaluated using gemini models, and other LLMs might react differently (addressed in limitations).
- Ablation studies have very low resolution (only 3 values per hyperparameter).
- The authors do not explicitly address the limitations of their work in the main paper, although they do so in the appendix.

---

> ### Author Rebuttal · Authors · 2025-07-30
>
> Thank you for your thoughtful and constructive review. We are delighted that you found our work's grounding in Cognitive Load Theory (CLT) to be "solid, highly interesting and has potential" and that you found our explanations and empirical evaluations to be clear and correct. Your suggestions are invaluable for improving the final version of our paper.
>
> Below, we address each of your points and outline the specific changes we will make in the final manuscript.
>
> **Response to Weakness 1 & Question 1: On the Motivation for Benchmark Selection**
>
> *What were the motivations or selection criteria for the evaluation benchmarks?*
>
> Our selection of **LiveBench** and **CommonGen-Hard** was a deliberate strategy to test our hypothesis under two distinct, complementary conditions of high cognitive load.
>
> 1.  **LiveBench as a SOTA, comprehensive "Real-World" Challenge:** We chose LiveBench because it is a well-known and SOTA benchmark for 2025 that contains comprehensive, real-world tasks, for example, **high school math competition, zebra puzzles, and word puzzles**. These tasks require a high cognitive load for human to solve. Its key advantages for our study are:
>     *   **Broad-Spectrum Evaluation:** It is built upon a diverse set of established and difficult benchmarks (e.g., Big-Bench Hard, AMPS [Math], IFEval [Instruction Following]) and real-world problems (e.g. International Mathematical Olympiad, Kaggle Competition), ensuring our framework is tested across a wide range of domains (e.g., math, reasoning, instruction following, data analysis and more).
>     *   **Fair and Rigorous Assessment:** Its frequent updates are specifically designed to minimize data contamination, ensuring that results reflect true reasoning capabilities rather than memorization, suitable to test SOTA models.
>     *   **High Cognitive Load Scenario:** The tasks are intentionally "hard"—even the most capable models struggle. This makes LiveBench an ideal proxy for the high cognitive load scenarios we aim to address, allowing us to probe the upper limits of a model's inherent problem-solving capabilities.
>
> 2.  **CommonGen-Hard as a Controlled, "Experimental" Challenge:** We selected CommonGen-Hard because it functions as a more systematic, experimental test of a core CLT principle: **information interactivity**. As a classic language task, its "Hard" version is naturally constructed to impose high cognitive load by forcing the model to integrate a few target concepts from a large pool of distractors. This provides a focused, controlled environment to validate that our architecture successfully manages precisely the kind of information interactivity predicted by CLT to cause performance degradation.
>
> **Action:** We will add a dedicated paragraph in **Section 5.1 (Evaluation Benchmarks)** to explicitly state this dual rationale.
>
> **Response to Weakness 2 & Question 2: On the Scope of Model Evaluation**
>
> *You only evaluate gemini models. Could you elaborate on that decision and how these results might or might not transfer to other model families?*
>
> Our core thesis is that *CoThinker* is a **model-agnostic framework that coordinates a model's inherent reasoning abilities to reduce cognitive load**. The framework's effectiveness does not depend on a model's internal parameters but rather on its ability to exhibit emergent social behaviors, such as following directed prompts, reacting to peer outputs, and synthesizing information. Therefore, any model that possesses these fundamental interaction capabilities should benefit from our method.
>
> To empirically validate this, we have conducted new experiments with models from other leading families. The results confirm our thesis, showing consistent performance gains. Notably, the improvement is more significant for models with stronger native reasoning, demonstrating that our framework enhances, rather than replaces, their capabilities.
>
> |Model|Method|Avg. Score (LiveBench)|Math Score|Reasoning Score|
> |:-|:-|:-:|:-:|:-:|
> |Gemini 2.5 Flash|CoThinker| **72.8**|**76.3**|**69.2**|
> ||DMAD|59.7|56.7|62.8|
> ||IO (Baseline)|45.1|59.3|31.0|
> |GPT-4.1-mini|CoThinker|**55.4**|**40.0**|**70.8**|
> ||DMAD|39.1|34.2|44.0|
> ||IO (Baseline)|37.4|34.0|40.8|
> |Qwen3-32B|CoThinker|**22.1**|**18.9**|**25.2**|
> ||DMAD|11.5|8.8|14.2|
> ||IO (Baseline)|11.7|3.4|20.0|
> |DeepSeek-R1 (qwen-8B)|CoThinker|**5.8**|**2.9**|**8.8**|
> ||DMAD|5.2|3.8|6.5|
> ||IO (Baseline)|2.3|1.9|2.8|
>
> **Action:** We will incorporate these new results and the expanded analysis into **Section 5.2** to robustly demonstrate the framework's generalizability.
>
> ---
>
> **Response to Weakness 3: On the Resolution of Ablation Studies**
>
> *Ablation studies have very low resolution (only 3 values per hyperparameter).*
>
> This is an accurate observation of our experimental design. The primary objective of the ablation studies was not to perform an exhaustive hyperparameter sweep, but to **empirically validate our CLT-based theory**. By showing that task-dependent optimal ranges exist—for instance, that Reasoning tasks benefit from a higher exploration rate (*β*) while Instruction Following does not (Figure 4)—we provide direct evidence for the cognitive trade-offs our framework is designed to manage. The resolution was sufficient to establish this principle.
>
> **Action:** We will refine the language in **Section 5.4 (Analysis of Ablation Studies)** to clarify that the study's scope was intentionally focused on validating these theoretical principles.
>
> ---
>
> **Response to Weakness 4 & Question 3: On the Placement of Limitations**
>
> *The limitations addressed in the appendix should be in the main paper.*
>
> We agree that this discussion is integral to the paper. To better contextualize our contributions and transparently outline the path for future research, the discussion of limitations and future directions belongs in the main text.
>
> **Action:** In the final version, we will integrate the **Limitations and Future Work** discussion from Appendix E into the main body of the paper as a new **Section 7**.
>
> ---
>
> **Response to the optional suggestion**
>
> *What were the motivations or selection criteria for the evaluation benchmarks? I can guess roughly from the rest of the paper, but it would be nice to have it explicitly estated, maybe also referencing one or two benchmarks that might seem suitable to inexpert eyes (this last bit is just a suggestion, so highly optional).*
>
> We appreciate your suggestion to clarify the motivations behind our benchmark selection and we add some examples of the actual datasets used in the benchmarks (See *Response to Weakness 1 & Question 1* above).
> Please let us know if this addresses your suggestion.

---

> > ### Comment · Reviewer_KZNb · 2025-08-01
> > **All concerns addressed**
> >
> > Thank you for addressing all my questions and concerns. I am satisfied with the responses and I think that the actions taken have improved the quality of the paper.

---

### Official Review · Reviewer_SVsA · 2025-06-30

**Clarity:** 3
**Significance:** 2
**Originality:** 2
**Rating:** 3
**Confidence:** 4

**Summary:**

This paper proposes a multi-agent in-context learning framework, CoThinker, to improve large language models' reasoning capabilities for complex tasks. The framework draws inspiration from human cognitive load theory (CLT) and collective intelligence observed in human teams. Empirical results indicate that CoThinker outperforms baseline agentic frameworks on highly complex reasoning tasks.

**Questions:**

1. How do the authors quantitatively or qualitatively verify that subdividing complex tasks into simpler subtasks indeed reduces cognitive load for LLMs?
2. What metrics or indicators were used to measure the cognitive load of each module within the architecture?
3. Could the authors clarify why the communication moderator relies solely on agent outputs from the previous round rather than considering the state of the transactive memory system?
4. How were the communication rules and topology decisions derived from cognitive load theory, and what justifications support these specific design choices?
5. How would the proposed framework generalize when implemented using state-of-the-art reasoning models such as Deepseek-r1, GPT-o3, or Gemini-2.5?
6. Given CoThinker’s sensitivity to hyperparameters (e.g., number of agents), how would one systematically tune these parameters and prompts for effective deployment in new, unseen domains?

**Ethical Concerns:**

["NO or VERY MINOR ethics concerns only"]

**Final Justification:**

The authors define explicit measurements of LLM working memory and evaluate them across tasks of varying complexity. The results provide strong evidence for the proposed analogy with cognitive load theory in explaining LLM reasoning capacity. The benchmark’s extension to newer base models further supports the validity of the framework. However, the rebuttal’s arguments for the method’s generalizability are less persuasive. Therefore, I am increasing both my quality score and final recommendation by 1 point.

**Limitations:**

Yes.

**Paper Formatting Concerns:**

No.

**Quality:**

3

**Strengths And Weaknesses:**

1. The theoretical analogy drawn in Section 3 between human cognitive load theory (CLT) and the limited reasoning capabilities of LLMs is overgeneralized. The concept of limited cognitive resources is imposed on LLMs based solely on observations of liomited reasoning capabilities similar to those in humans. No formal proof or empirical evidence is provided to link CLT to specific internal representations or mechanisms within LLMs. For instance, it is unclear whether the context window of in-context learning or certain intermediate neuron activations should be considered the "working memory" of an LLM.
2. The proposed implementation of collective intelligence is distinct from how actual human teams function. Specifically, submodules like the "transactive memory system" and "communication moderator" are not actual entities in human teams but rather are concepts that emerge from the interactions (e.g., negotiation) among multiple autonomous agents. It is not clear from a theoretical standpoint how the proposed collective intelligence analogy differs from existing Multi-Agent Systems (MAS) that employ similar mechanisms like communication, shared memory, and theory of mind. Consequently, the paper's first claimed contribution of drawing a “strong” analogy between CLT and the limited reasoning capabilities of LLMs is not convincing.
3. The key assumption of the proposed CoThinker architecture is that LLMs can better handle sub-tasks divided from the original complex task due to a lower cognitive load. However, this is not verified quantitatively or qualitatively in the paper.
4. The retrieval process shown in Figure 2 is not explained in the main text. It appears that the communication moderator only considers an agent’s output from the last round without taking the state of the transactive memory system into account. This design decision seems counter-intuitive and requires further explanation, as the main purpose of maintaining shared memory (i.e., "who knows what") is to guide effective information sharing among team members.
5. The communication topology and algorithm are determined by a set of pre-defined rules. More explanation is needed to justify the rationale for generating these rules and their connection to CLT.
6. The benchmark baseline section is missing state-of-the-art models trained with native reasoning capabilities (e.g., DeepSeek-R1, GPT-o3, Gemini 2.5 pro). The value of the proposed agentic framework with a fixed workflow is less clear given new training methods based on reinforcement learning and internal reasoning chains.
7. As shown in the ablation studies, CoThinker is sensitive to hyperparameters, such as the number of agents. This raises the question of how to tune these hyperparameters and prompts when applying the proposed framework to a new, unseen domain.

---

> ### Author Rebuttal · Authors · 2025-07-30
>
> Thank you for your detailed review. After carefully considering your comments, **we clarify some facts on Sec. 3: Our analogy isn't only based on observations but also grounded in theoretical and empirical evidence.** It begins with a foundational analogy: human brain can only attend to a limited amount of information at once, a feature known as *working memory* (*Sec. 3.1*). The architecture of LLM, due to its inherent attention mechanism, similarly restricts the amount of information it can focus on at once, where we refer this LLM feature as LLM's *working memory* (*Response 1* for theoretical justification).
>
> We then show how Cognitive Load Theory (CLT) effectively manages WM feature (Limited focused information capacity), thereby guiding human interaction to enhance efficient group collaboration in solving complex problems involving diverse knowledge integration. CLT provides a *design principles* to manage effective information within limited WM (*section 3.2*).
>
> Since MAS simulates human interactions to solve complex problems involving diverse knowledge integration, the *design principles* extracted from human CLT can be **reused** to inform the design of MAS. These *design principles* guide our framework's setting of Communication Moderator, Transactive Memory System, and Parallel Thinking Modules, which differentiates us from existing MAS (*Response 2* for details).
>
> **Response to W1, W2 & Q1, Q2:** *The theoretical analogy...is overgeneralized...No formal proof or empirical evidence...key assumption...not verified...subdividing tasks reduces cognitive load? What metrics...*
>
> We address these concerns through a unified logic chain that establishes WM in LLM, validates the CLT analogy, and proves our core assumption with direct empirical evidence.
>
> **Step 1: Establish WM in LLM & Measure Cognitive Load**
> We identify the key *characteristics* of WM, then dual *characteristics* in LLM, finally empirical evidence.
>
> 1. Key Characteristics of WM:
> By definition, WM is a feature that temporarily holds and processes information simultaneously and cognitive load is the *attention* required to handle information within WM, which determines the *easiness* of task completion [1]. Thus, the key characteristics to quantify WM are "attention on interactivity of information" and "easiness of task completion". LLM *attention mechanism* & *perplexity* naturally become dual characteristics quantifying these aspects.
>
> 1. Pre-experiment 1: Attention as a proxy for WM
> $$
> \text{Attention Entropy} = -\sum_{i=1}^N a_i \log a_i
> $$
> measures attention distribution diversity. Higher entropy indicates uniform distribution, suggesting the model considers multiple aspects of the input, corresponding to higher cognitive load [7].
>
> We use Q&A pairs (AMPS-Hard) w/ 4-level difficulty, then control length of input for fair comparison and measure attention entropy on the answers:
>
> |Task Complexity|Attention Entropy (No Reasoning)|Attention Entropy (With Reasoning)|
> :-|:-:|:-:
> 1|4.442|4.439
> 2|4.796|4.726
> 3|5.043|4.937
> 4|6.101|5.920
>
> Attention entropy increases with complexity, indicating the model have to use more information pieces to process the task, indicating higher cognitive load. Presence of reasoning steps lowers attention entropy, indicating that reasoning helps the model to "offload" partial cognitive load by focusing on key informations.
>
> 1. Pre-experiment 2: Perplexity as a proxy for WM
> $$
> \text{Perplexity}=\exp\left(-\frac{1}{N}\sum_{i=1}^{N}\log p(s_i)\right)
> $$
> measures certainty of the model's generation, which is the "easiness" of task completion [6].
>
> We choose Q&A pairs (FLASK) w/ 5-level guiding instructions complexity, then measure the perplexity on answers:
>
> Instruction Complexity|Perplexity (hard)|Perplexity (easy)
> :--|:-:|:-:
> 1|120.5013|3.3661
> 2|88.9650|3.4227
> 3|85.3538|3.4522
> 4|92.4836|3.4572
> 5|100.7143|3.4603
>
> For hard tasks, the perplexity decreases then increases, showing that instructions help LLM to focus on the task up to a certain point, and then become additional burdens on the model's cognitive load; While for easy tasks, the perplexity increases, the instructions showing no help. This measurement of "easiness" aligns with CLT (*Sec. 3.2*).
>
> **Step 2: Validate CLT Analogy**
> Similar to CLT initially grounded in behavioral observations & experiments [1], our analogy uses observed LLM limitations [2,3] & previous "WM capacity" experiments [4,5], which consists with cognitive science research. Furthermore, with established **metrics** above, we show LLM exhibit the same fundamental WM limitations.
>
> **Step 3: Subtasking Reduces Cognitive Load & Show CLT Principles Work**
> Using established metrics, CoThinker's "subtasking" means each agent attends to smaller, focused info sets, reflected by lower Attention Entropy and Perplexity, providing quantitative evidence of reduced CL. This causal mechanism directly accounts for performance gains in ablation studies (Fig 4): System performs better only on high CL tasks, where subtasking is helpful.
>
> **Clarification on Module-Level Measurement**: Our modules are functional tools/algorithms, not cognitive agents. CL measurement only makes sense for individual agents, not the tools themselves.
>
> ---
> **Clarification on W2:** *...implementation...is distinct from actual human teams...differs from existing MAS...*
>
> We clarify that *Transactive Memory System (TMS)* and managing group cognitive load (*Communication Moderator (CM)* ) are established theoretical concept from human teams, describes as *human emergent behavior* in social and cognitive psychology [8,9]. These aren't engineering choices but rather *design principles* derived from CLT. Our framework explicitly grounds in CLT, actively using *design principles* to operationalize these emergent behaviors into modules to guide the MAS effectively. Each design choice follows CLT principles, distinguishing us from existing MAS. Please refer to **Clarification 4 & 5** for details on *TMS* & *CM*.
>
> ---
> **Clarification to W6 & Q5:** *Benchmark baseline missing...*
>
> Our framework doesn't compete with a model's native reasoning but to **coordinate it effectively**. As long as the model possesses certain emergent social behaviors, our framework can enhance its performance by reducing cognitive load generally. Our approach is **model-agnostic**, coordinating a model's capabilities rather than changing them. FYI, results from other model families:
>
> Model|Method|Avg. Score (LiveBench)|Math|Reasoning
> :-|:-|:-:|:-:|:-:
> Gemini 2.5 Flash|CoThinker| **72.8**|**76.3**|**69.2**
> ||DMAD|59.7|56.7|62.8
> ||IO|45.1|59.3|31.0
> GPT-4.1-mini|CoThinker|**55.4**|**40.0**|**70.8**
> ||DMAD|39.1|34.2|44.0
> ||IO|37.4|34.0|40.8
> Qwen3-32B|CoThinker|**22.1**|**18.9**|**25.2**
> ||DMAD|11.5|8.8|14.2
> ||IO|11.7|3.4|20.0
> DeepSeek-R1 (qwen-8B)|CoThinker|**5.8**|**2.9**|**8.8**
> ||DMAD|5.2|3.8|6.5
> ||IO|2.3|1.9|2.8
>
> Results on these SOTAs show greater improvement. Our framework enhances reasoning efficiency. DeepSeek-R1 model struggled w/ mid-level math problems within 4k output context window; CoThinker enabled it to find solutions.
>
> ---
> **Clarification on W4 & Q3:** *The retrieval process... not explained...*
>
> Agent receives input from both TMS & CM. Two-channel design is designed to manage cognitive load:
> 1. TMS provides **high-level, synthesized, long-term context** (group progress, consensus), giving agent strategic awareness over rounds w/ low cognitive burden.
> 2. CM provides **specific, low-volume, current peer outputs** for immediate reaction & refinement.
>
> "Who knows what" can be seen as long-term info for peers to quickly build consensus, guiding agent's current understanding of its peers. (e.g. How much should I rely on this peer? What detail this peer ignores so I should explore?)
>
> ---
> **Clarification on W5 & Q4:** *Communication topology & algorithm determined by pre-defined rules...*
>
> Rules for communication algorithm are direct operationalization of CLT principles:
>
> 1. **Rule 1: Fixed In-Degree (N):** Directly enforces CLT principle of limited WM. By capping number of incoming peer messages, we prevent any single agent from high extraneous cognitive load from info overload.
> 2. **Rule 2: Adaptive Rewiring ($\beta$):** This rule is designed to create a adaptive network by managing the trade-off between exploiting similar ideas (lower $\beta$ reduces extraneous load) and exploring diverse ones (high $\beta$ distributes intrinsic load). This is principled approach supported by network science [10,11].
>
> ---
> **Response to W7 & Q6:** *CoThinker is sensitive to hyperparameters...*
>
> We view sensitivity not as weakness, but as **empirical confirmation of our thesis**: parameters directly correspond to cognitive trade-offs predicted by CLT. Existence of optimal, task-dependent range is what our theory expects. Future work involves estimating task's cognitive load at runtime & dynamically adjusting hyperparameters, creating self-regulating system.
>
> # References
> [1] Sweller, J. "Cognitive load theory." Psychology of Learning and Motivation, 2011.
>
> [2] Liang et al. "Encouraging Divergent Thinking in Large Language Models through Multi-Agent Debate." EMNLP, 2024.
>
> [3] Huang et al. "Large language models cannot self-correct reasoning yet." ICLR, 2024.
>
> [4] Gong et al. "Working memory capacity of ChatGPT: An empirical study." AAAI, 2024.
>
> [5] Zhang et al. "Working memory identifies reasoning limits in language models." EMNLP, 2024.
>
> [6] Xiong et al. "Can LLMs Express Their Uncertainty? " ICLR, 2024.
>
> [7] Zhang et al. "Attention Entropy is a Key Factor." ACL, 2025.
>
> [8] Zhang et al. "Exploring Collaboration Mechanisms for LLM Agents: A Social Psychology View." ACL, 2024.
>
> [9] Kirschner et al. "A cognitive load approach to collaborative learning." Educational Psychology Review, 2009.
>
> [10] Almaatouq et al. "Adaptive social networks promote the wisdom of crowds." PNAS, 2020.
>
> [11] Rand et al. "Dynamic social networks promote cooperation in experiments with humans." PNAS, 2011

---

> > ### Author Response · Authors · 2025-08-02
> >
> > Dear Reviewer SVsA,
> >
> > Thank you again for your valuable time and detailed review. We have posted a rebuttal where we sought to address your concerns with new empirical evidence and further clarifications.
> >
> > We would be grateful to know if our response has helped clarify these points and would be happy to discuss any remaining questions you may have.
> >
> > Thank you once more for your insights.
> >
> > Best regards,

---

> > ### Comment · Reviewer_SVsA · 2025-08-04
> >
> > Thank you for the thorough rebuttal and the additional experiments. These address most of my earlier concerns, so I am raising my score to 3.

---

> > > ### Author Response · Authors · 2025-08-05
> > >
> > > Dear Reviewer SVsA,
> > >
> > > Thank you for taking the time to review our rebuttal and for acknowledging that our responses and new experiments have addressed most of your concerns. We are grateful for your feedback and for raising our score.
> > >
> > > All the best,

---

### Note · Authors · 2025-08-13

We sincerely thank all reviewers for their constructive feedback. Our rebuttal addressed their key concerns, with positive responses from all reviewers: SVsA and bai9 raised scores, KZNb confirmed all concerns addressed, and evwp found our responses more reasonable.

**Recognized Strengths**
Reviewers acknowledged our work's: Novel CLT-grounded approach as "solid, highly interesting" [KZNb]; Principled design with CLT-motivated modules distinguishing CoThinker from existing MAS [SVsA,bai9]; Insightful network analysis and ablation studies [evwp,bai9]; Clear presentation [KZNb].

**Key Clarifications in Rebuttal**
We provided justification addressing two primary concerns:

1. Theoretical Foundation (SVsA, evwp): We first clarified the validity of our CLT-LLM analogy by establishing the architectural foundation and methodology through observation of LLM behavioral patterns paralleling human cognitive limitations. We then provided novel pre-experiments showing attention entropy and perplexity as quantitative proxies for cognitive load, with empirical evidence, further consolidating our theoretical framework
2. Model-Agnostic Generalizability: We clarified CoThinker's model-agnostic nature stems from using emergent social behaviors observed across most current LLM. We then provide more experiments across diverse model families including reasoning models (GPT, Qwen, DeepSeek, Gemini), finding even larger enhancements in stronger models, confirming our framework works generically

**Paper Enhancements**
1. Enhanced Transparency and Details (evwp, bai9): Added information flow diagrams, architectural comparisons with AutoGen, explicit benchmark selection rationale, cross-model evaluation results, and agent specialization ablation studies
2. Improved Structure and Clarity: Moved limitations to main paper Section 7, specified appendix references, added granular performance analysis, detailed TMS and Communication Moderator design rationales, clarified cognitive offloading mechanisms

**Key Contribution**
CoThinker establishes a foundation analogy between human and LLM working memory, thus enabling using human CLT principles to guide multi-agent LLM coordination. By demonstrating consistent performance gains across diverse LLMs through cognitive load management, we bridge AI and cognitive science in a principled way. Our framework's plug-and-play cognitive principles complement existing MAS, advancing collaborative AI systems through a novel cognitive science lens.

---

### Decision · Program_Chairs · 2025-09-17

**Decision:**

Reject

**Comment:**

The paper presents a multi-agent in-context learning framework named CoThinker to improve LLMs reasoning capabilities, arguing LLMs suffer from a limited cognitive load capacity on complex tasks. CoThinker is inspired by cognitive load theory (CLT) and collective intelligence observed in human teams.  The framework distributes cognitive load among specialized LLM agents through shared memory and structured communication.  Empirical results show that the framework improves performance and reasoning efficiency on complex problem-solving tasks and fabricated high cognitive load scenarios.

This is a borderline paper with mixed reviews. The reviewers appreciate the novelty in grounding LLM capability into the cognitive load theory, and the novelty of the CLT-inspired solution.However, the reviewers shared concerns about limited experimental setting and unclear theoretical foundation of CLT-LLM analogy. After rebuttal and discussion period, two reviewers increased their ratings. Reviewer SVsA has a remaining concern regarding the gap between CLT and the proposed framework, as similar architecture could be proposed without resorting to CLT: "Compare to CLT, the authors might gain more valuable insights by drawing from research in industrial and organizational psychology or team science when designing agent networks." Thus, the recommendation is rejection. The authors are encouraged to clarify the connection between CLT and the proposed framework in future version.  A deeper ablation study would also be helpful to validate the CLT-inspired design such as by removing some component for potential degraded performance.